# Marangoni-driven deterministic formation of softer, hollow microstructures for sensitivity-enhanced tactile system

Wennan Xiong [1,2], Fan Zhang [1,2] ✉, Shiyuan Qu [1,2], Liting Yin [1,2], Kan Li[1,2] & YongAn Huang [1,2] ✉

Microengineering the dielectric layers with three-dimensional microstructures has proven effective in enhancing the sensitivity of flexible pressure sensors. However, the widely employed geometrical designs of solid microstructures exhibit limited sensitivity over a wide range of pressures due to their inherent but undesired structural compressibility. Here, a Marangoni-driven deterministic formation approach is proposed for fabricating hollow microstructures, allowing for greater deformation while retarding structural stiffening during compression. Fluid convective deposition enables solute particles to reassemble in template microstructures, controlling the interior cavity with a void ratio exceeding 90%. The hollow micro-pyramid sensor exhibits a 10-fold sensitivity improvement across wider pressure ranges over the pressure sensor utilizing solid micro-pyramids, and an ultra-low detect limit of 0.21 Pa. With the advantages of facilitation, scalability, and large-area compatibility, such an approach for hollow microstructures can be expanded to other sensor types for superior performance and has considerable potential in robotic tactile and epidermal devices.

Three-dimensional (3D) microstructures have attracted much interest and shown key applications in biomedical monitoring[1–5], human-machine interfaces[6–9], robotic tactile[10–14], and microfliers[15] due to their great superiorities in softness, lightweight, conformality on 3D surfaces and compressibility[16–20]. Flexible capacitive pressure sensors (CPSs), consisting of two parallel plates sandwiched with a soft dielectric film between them, have demonstrated many attractive advantages of simple device construction, high sensitivity, and fast responding speed[21–25]. The 3D mechanical design of the structured dielectric layer plays a crucial role in determining sensor performances, including sensitivity, which is a vital feature for pressure sensors as it determines the capacities of perceiving subtle pressures and acquiring more details, such as gentle touch[26] and pulse detection[27,28]. Since the pyramid microstructured dielectric layer is fabricated by Bao et al., replacing the viscoelastic and incompressible

blanket film, to achieve a high sensitivity of 0.55 kPa$^{-1}$ (<2 kPa)[29], adding voids in the dielectric layer through microstructural engineering have been widely explored as an effective strategy for high sensitivity[30–32]. Various attempts have been made to develop different 3D microstructures, such as micro-pillar arrays[33], micro-domes[6,34], micro-stripes[29,35], and plant-based irregular structures[22,36], through replicating from a premade mold[37], and electrically responsive self-growing strategy[38]. The air voids between protruding microstructures, replacing solid constructions, improve the compressibility of the intermediate dielectric layer, which significantly facilitates the heightened sensitivity of flexible pressure sensors[39] (including but not limited to capacitive[23], piezoresistive[40], iontronic[41] sensing mechanisms). Indeed, a key research direction has been clarified to produce microstructures with greater height-to-width aspect ratios, which are easier to destabilize and possess more room to be compressed, through

[1]State Key Laboratory of Intelligent Manufacturing Equipment and Technology, Huazhong University of Science and Technology, Wuhan 430074, P.R. China. [2]Flexible Electronics Research Center, Huazhong University of Science and Technology, Wuhan 430074, P.R. China. ✉e-mail: fanzhang@hust.edu.cn; yahuang@hust.edu.cn

advanced techniques such as photolithography[33], laser ablation[42,43], magnetically induced growth[44,45]. By using a laser-engraved acrylic mold, a micro-needle structure with an aspect ratio of up to 11.8 has been created, resulting in a significantly enhanced sensitivity[43]. But these solid microstructures primarily deform by self-mutual extrusion when subjected to a pressure load[6,22,46], and have increasingly exhibited distinct fatigue in sensitivity further improvement due to their inherently high compression modulus. Moreover, the high sensitivity caused by solid microstructures is limited to a low-pressure regime, attributing to their structural stiffening upon increasing pressure as previously reported[22,23,29,43]. To achieve an increasingly high-aspect ratio, further development of advanced or complex technological processes is necessary.

Further work has demonstrated combining multiple approaches, such as introducing micropores to 3D microstructures, to help further enhance compressibility. A porous micro-pyramid structure is developed by dissolving polystyrene beads pre-coated on the pyramid-shaped polydimethylsiloxane (PDMS) surface in toluene[21]. The presence of micropores makes the structure more compressible, yielding a greatly improved sensitivity compared to solid pyramids. However, while such a combined approach proves effective at low pressures, the advantages over solid pyramid structures diminish at higher pressures due to the low void ratio that micropores distribute only on the surface[30]. In comparison, hollow structures, generally containing an interior cavity, will be softer and provide more space for compression under high pressure, which denotes a promising trend for pressure sensors to address the longstanding challenge of high sensitivity over a broad pressure range. Previous work has reported on the preparation of hollow micro-domes, nevertheless, the process of which is cumbersome and challenging to fabricate hollow structured arrays with micrometer scale[47].

Here, a Marangoni-driven deterministic formation approach is harnessed to fabricate a soft film featuring hollow structured micro-arrays. During the drying process of the polymer solution, fluid convective deposition allows the remaining solute particles to reassemble and distribute in the mold pit for cavity formation within microstructures. The interior cavity can be controlled to attain a high void ratio of over 90%, of which the geometric profiles can be precisely predicted using the model grounded in Gaussian curves with root mean square errors of below 2 μm. Compared to conventional solid structures, the inner cavity of hollow microstructures experiences the majority of compressive deformation during compression, resulting in exceptional compressibility and superior resistance to structural stiffening. The results indicate that, despite having a low aspect ratio of 0.7, the hollow micro-pyramid (HMP) enhanced CPS exhibits higher sensitivities over wider pressure ranges (i.e., from 3.45 kPa$^{-1}$ within 0−0.6 kPa to 0.13 kPa$^{-1}$ within 10−40 kPa) than the sensors utilizing the high-aspect-ratio microstructures like needle shapes. Even though the sensitivity of our sensor still decays with compression, the decline is relatively gentle, and it maintains higher levels within the same ranges. An exact comparison demonstrates that the HMP-enhanced sensor has yielded a tenfold improvement in sensitivity and a wider range over the sensor based on solid micro-pyramids of identical dimensions. Additional advantages are also found, such as a fast response time of 16.8 ms, an ultra-low limit of detection (LoD) of 0.21 Pa, as well as exceptional stability even after 10,000 cycles. Finally, the HMP-enhanced sensor is verified by integrating with the robotic electronic skin for tactile perception, which always involves detecting subtle pressure changes under significant pre-contact pressures, to construct an automatic robotic pulse-diagnosis system combined with the vision-aided location algorithm. The proposed fabrication process for hollow microstructures is also compatible with the prevalent replica-molding technology for solid microstructures, which is simple, scalable, and large-area compatible. We believe that such a structural design and fabrication, serving as a fundamental strategy, can be extended to other types of sensors for high performance.

## Results

### Formation of the HMP

Replica molding, one type of soft lithography, is a prevalent and effective method for creating soft films featuring microstructured surfaces. The process starts with a premade original template coated with a polymer solution, which determines the final shape of the 3D microstructure. The formation of the interior cavity within the 3D microstructure depends on the deposition and distribution of the remaining solutes in the template pit, and the fabrication process of the HMP film is illustrated in Fig. 1a. The polymer solution, consisting of polyvinyl alcohol (PVA), magnesium chloride (MgCl$_2$), and deionized water, is poured onto a prepared inverted pyramid mold. Different from the commonly used PDMS or Ecoflex elastomers relying on the cross-linking reaction of two components without any mass loss, the water of this polymer solution will constantly evaporate to form solid films with an apparent volume decrease and induced liquid flow. After the polymer solution drying, the major components of the HMP structured film are the residual solutes (PVA and MgCl$_2$).

Firstly, the polymer solution between each inverted pyramidal pit evaporates to form a dried film, thereby establishing a solid-liquid interface, as depicted in Fig. 1a. As evaporation progresses, the interface gradually moves towards the center of the inverted pyramidal pit, eventually being pinned at the pit edge[48]. The remaining solution within the template pit determines the final morphology of the interior cavity. Due to a larger surface-volume ratio, the higher evaporation rate appears intrinsically near the solid-liquid interface, which results in an outward capillary flow transporting the solute particles from the center towards the edge of the inverted pyramidal pit where deposition occurs[48–50]. During the evaporation phase, when an external heat source like a hot plate is applied to the mold, the liquid convection and evaporation processes become more intensified[51]. Meanwhile, a temperature gradient from the center to the edge of the pit will be developed intrinsically along the air-liquid interface, attributed to the difference in thermal conductivity between the PVA/MgCl$_2$ solution and the PDMS mold. The surface tension of the polymer solution decreases as the temperature of the air-liquid interface rises, causing a Marangoni flow from the center of the pit[52]. Hence, the continuous evaporation-induced convective deposition promotes the formation of interior cavities within 3D microstructures, and this effect is enhanced at elevated temperatures.

After completely drying and peeling from the mold, a soft dielectric film featuring arrayed HMPs is readily obtained, as displayed in the tilt-view scanning electron microscopy (SEM) images of Fig. 1b, c and Supplementary Fig. 1. The top surface of the structured film retains the shape of the inverted pyramid template, while the bottom surface exhibits a uniform concave configuration. The cross-sectional SEM image (Fig. 1d) also reveals the cavity within a pyramid, highlighted by a white dashed line in the enlarged view. These results demonstrate that the proposed approach is effective in forming hollow pyramid microstructures. The surface topographies of the interior cavities of the micro-pyramids are compared at five areas (center and four corners) of the microstructured array (a size of 12.5 mm × 12.5 mm) by using laser scanning confocal microscopy, indicating consistent shapes and dimensions of the hollow micro-pyramid structures across various locations (Supplementary Fig. 2). The replica molding incorporating with the evaporation-induced microfluidic formation provides a versatile approach to create interior cavities at micrometer scale and can also be applied to develop a wide variety of hollow microstructures with various shapes, no requiring additional processing steps. Supplementary Fig. 3 shows SEM images of the hollow cylinder, cuboid, and dome microstructures that are prepared with the various template shapes. The simulation results of compressive strain versus applied pressure (Fig. 1e) indicate that hollow microstructures (e.g., pyramid, bubble) exhibit superior structural compressibility,

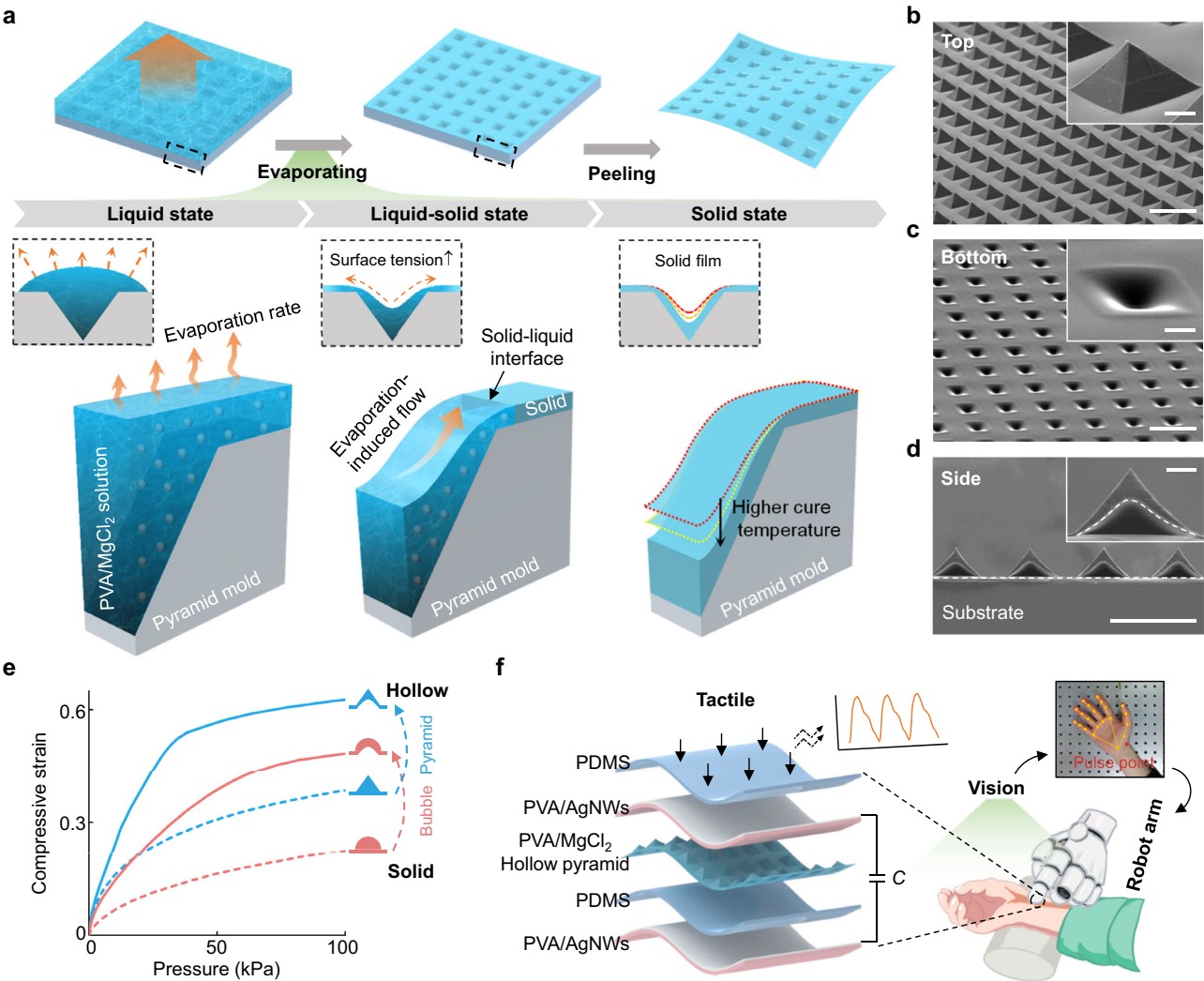

**Fig. 1 | Concept of the hollow microstructure formation via the Marangoni-driven microfluidic assembly method. a** Schematic illustration of the formation process of the hollow micro-pyramid (HMP) structure by the evaporation-induced flow. **b**–**d** SEM images of the HMP array with **b** top view, **c** bottom view, and **d** cross-sectional view. Scale bars, 200 μm. The insets show the zoom-in micro-pyramid structure. Scale bar, 30 μm. **e** Numerical simulated compressive strain responses of the hollow and solid microstructures under different levels of applied pressure. **f** Schematic illustration of the construction and application of a capacitive pressure sensor with the HMP array.

such as greater deformation for high sensitivity while retarding structural stiffening for a wide sensing range, compared to solid microstructures. The hollow micro-pyramid serves as the dielectric layer in the construction of a CPS for robotic tactile, which demonstrates a system-level application of an autonomous pulse-diagnosis robot with a vision-aided location algorithm (Fig. 1f). This CPS consists of five layers, a bottom PVA film with silver nanowires (AgNWs) electrode as the substrate, a PDMS insulation layer, a hollow micro-pyramid array (PVA/MgCl$_2$) employed as the dielectric layer, a top PVA film with AgNWs electrode, and a PDMS encapsulation layer.

## Characterizations of the HMP

Optimizing the process conditions of solvent casting is essential for the fabrication of pressure sensors where structural compressibility is closely related to the interior cavity. During the solvent evaporation phase, the formation of hollow microstructure is intimately connected to the deposition and distribution of remaining solutes, directly determining the final morphology of the cavity, such as cavity height ($h$), total height ($H$), film thickness ($t$), and full width at half maximum ($b$), as depicted in Fig. 2a. The base width ($B$) is determined as the bottom length of the outer surface of a micro-pyramid structure,

which is the same as that of a solid pyramid. A high hollow-to-total height ratio ($\alpha = h/H$) is desirable, as it signifies increased interior voids within the micro-pyramid, which is essential for enhancing structural compressibility. These geometric parameters are closely related to the following fabrication conditions: solution properties (e.g., polymer concentration, deposition volume), pyramid size, and temperature-induced fluid flow, which are explored to derive the expression for the cavity profile. The film thickness coefficient ($\lambda = t/T$) is defined as a key feature of the polymer solution to evaluate the possibilities of fabricating hollow microstructures, where $t$ is the film thickness between the pits and $T$ is the solution height as illustrated in Fig. 2b. The solution height $T$ can be calculated by $T = V/A$, where $A$ is the area of the mold. The thickness coefficient represents the proportion of remaining solutes (PVA and MgCl$_2$) following the evaporation of the solution. A low thickness coefficient is conducive to producing hollow microstructures with large height ratios or small base widths. The parameter $t$ is expected to increase linearly with the solution height for the solutions with different doping ratios of MgCl$_2$·6H$_2$O ($wt_{MgCl_2 \cdot 6H_2O}$), indicating that the solution volume has a linear impact on the formation of hollow microstructures (Supplementary Fig. 4). The calculated thickness coefficients of polymer solutions are presented in Fig. 2b,

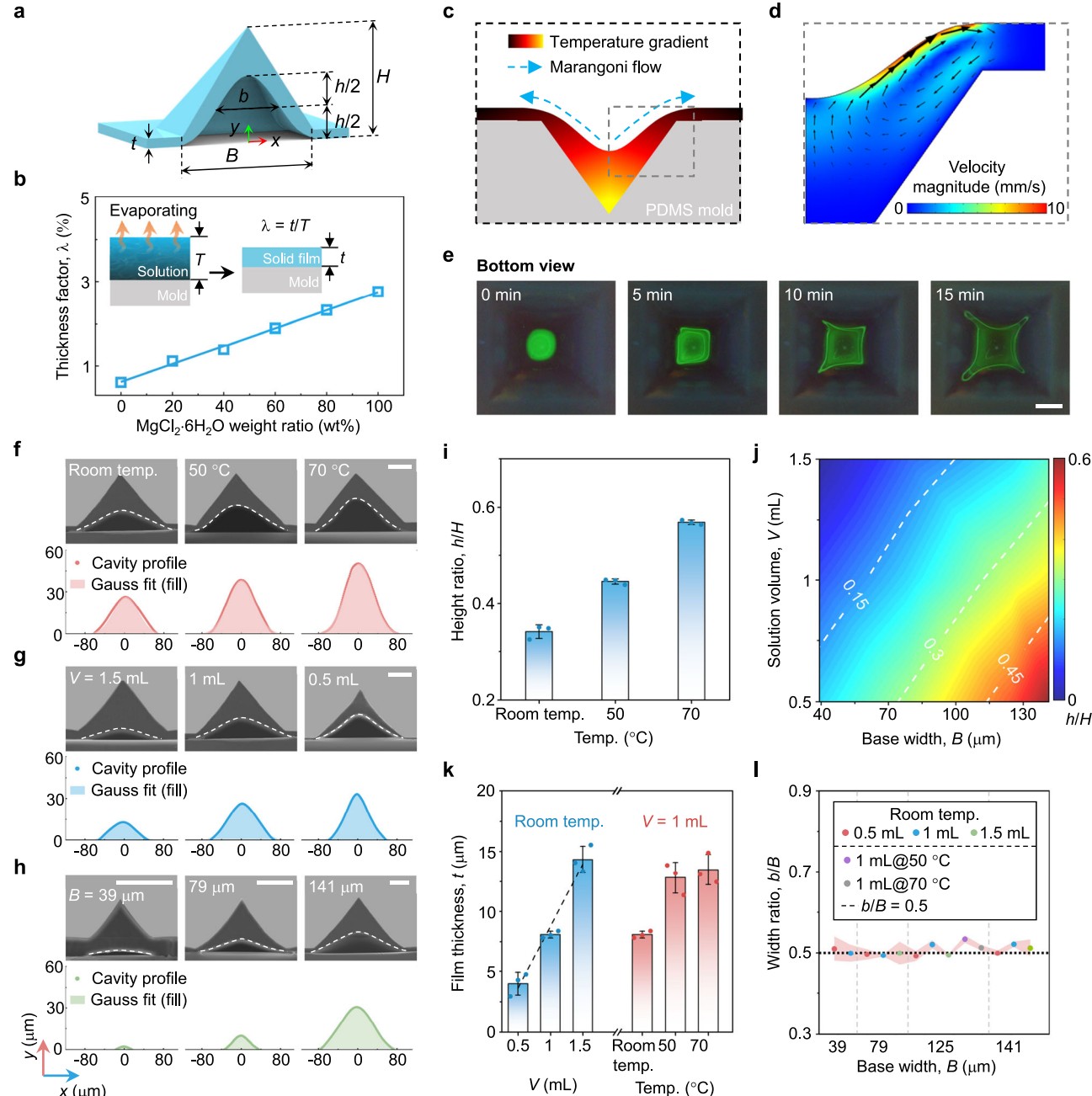

**Fig. 2 | Characterizations of the HMP structure. a** Schematic illustration of geometric parameters of a single HMP, including base width $B$, film thickness $t$, cavity height $h$, total height $H$, full width at half maximum $b$. **b** Thickness factor of the polymer solution with varying doping ratios of $MgCl_2 \cdot 6H_2O$. **c** Depiction of temperature gradient-induced Marangoni flow within the polymer solution under heating conditions. **d** Simulation result of flow velocity distribution of polymer solution with a heating temperature of 70 °C (at the bottom of PDMS mold). **e** Visualization of Marangoni flow in the solution with green fluorescent particles, where the surface position of the solution is about half of the height of the micro pyramid. Scale bar, 2 mm. **f–h** Cross-sectional SEM images of the HMPs formed under different conditions of **f** curing temperatures, **g** solution volumes, and **h** base widths, and extracted geometric profiles (white dashed lines) of the cavities fitted with Gaussian curves. **i–l** Geometric parameters of hollow micro-pyramids in terms of curing temperature, solution volume, and base width. **i** Hollow-to-total height ratio ($h/H$) versus curing temperature. Error bars show s.d., $n = 3$. **j** Contour plot of $h/H$ versus solution volume ($V$) and base width ($B$). **k** Film thickness ($t$) versus solution volume and curing temperature, respectively. Error bars show s.d., $n = 3$. **l** Width ratio ($b/B$) versus base width.

and a linear expression can be used to estimate the corresponding relationship,

$$\lambda = (21.245 wt_{MgCl_2 \cdot 6H_2O} + 6.2) \times 10^{-3}. \quad (1)$$

At a weight ratio of 20%, the solution has a low thickness coefficient of -1%.

Figure 2c demonstrates the elevated mold temperature and the corresponding temperature gradient-induced outward Marangoni flow at the air-liquid interface. A temperature gradient appears when the pyramidal mold is baked, resulting in an outward Marangoni flow. The numerical result is shown in Fig. 2d with a heating temperature of 70 °C and an ambient temperature of 20 °C. An apparent outward Marangoni flow can be observed near the liquid-air interface, facilitating the supply of solute particles to the deposition far from the

center of the pyramidal pit. Furthermore, the fluid flow is visualized by the fluorescein-tracing method. When the solution evaporates to half the height of the pyramid, a dilute green fluorescent droplet is added to the polymer solution (Supplementary Fig. 5). A fast backflow can be observed in Fig. 2e that carries the fluorescent particles to the solid-liquid interface, which is advantageous for the enlargement of the cavity (Fig. 2f). As increasing the curing temperature, and the height ratio also increases linearly (Fig. 2i). At room temperature, the height ratio is 0.34 when $B = 125\,\mu m$ and $V = 1\,mL$. When the temperature increases to 50 and 70 °C, the height ratio grows to 0.45 and 0.57, respectively.

The cross-sectional SEM images of hollow pyramid microstructures formed under different levels of solution volumes and base widths are shown in Fig. 2g, h, respectively. A greater base width provides increased space for solute particle deposition, while a smaller solution volume means fewer remaining solutes, which can produce a more expansive cavity (see more cross-sectional SEM results under different base widths of 39, 79, 125, 141 μm and solution volumes of 0.5, 1, and 1.5 mL in Supplementary Fig. 6). The statistical results of hollow-to-total height ratios in terms of solution volumes and base widths are shown in Fig. 2j. The maximum height ratio, appearing in the lower right corner, is equal to 0.6 for $B = 141\,\mu m$ and $V = 0.5\,mL$. The height ratio is zero, indicating a solid pyramid microstructure, for $B = 39\,\mu m$ and $V = 1.5\,mL$. Notably, the height ratio ($h/H$) shows a significant linear increase with the base width of the micro pyramid ($h/H \propto B$) and a decrease with the solution volume ($h/H \propto -V$), as shown in Supplementary Fig. 7.

Consequently, the temperature-induced growth coefficient (i.e., the slope of height ratio-curing temperature curve) of 0.11 per 20 °C is approximate to the volume-induced growth coefficient of 0.1 per 0.5 mL. Both of these factors can enable the height ratio to expand to ~0.6, indicating their equal importance in the formation of hollow microstructures. Nevertheless, producing a hollow microstructure with a high height ratio by minimizing the solution volume will yield an ultrathin film thickness. For instance, when the solution volume decreases from 1 to 0.5 mL, resulting in a height ratio increase of 0.1, the film thickness declines from 8 μm to a mere 4 μm (Fig. 2k). The consequent small thickness will pose a new challenge in peeling such an ultrathin film from a template. The temperature gradient-induced flow compensates for the shortage of changing the solution volume to produce a hollow microstructured dielectric film with a larger height ratio and film thickness, as shown in Fig. 2e. When the heat temperature increases to 50 °C, causing the same improvement in height ratio, the film thickness $t$ increases to 13 μm, which is more conducive to successful demolding.

To further investigate the morphology of hollow pyramids, geometric profiles of the cavities formed under different levels of solution volumes, curing temperatures, and base widths are extracted from the SEM images and then fitted with Gaussian curves, as presented in Fig. 2f–h, respectively (see more curve-fitting results in Supplementary Fig. 8). The curve equation can be given as

$$y = h e^{-\frac{(x-u)^2}{2v^2}} \qquad (2)$$

where $u$ is equal to 0 for the symmetrical structure, $v$ is related to the full width at the half of the peak ($b$), and is described by $v = b/(2\sqrt{2\ln 2})$. The cavity height can be expressed by the formula $h = \alpha(B\tan\theta/2 + t)$, derived from the geometric relationship of pyramids. The template micro-pyramid pits are firstly created by etching (100) silicon wafers using potassium hydroxide (KOH), producing a fixed side angle ($\theta$) of 54.7° for the pyramid[53]. The root mean square errors (RMSEs) are calculated and plotted in Supplementary Fig. 9, where the maximum RMSE is below 2 μm. The experimental results are in good agreement with theoretical predictions [by Eq. (1)]. Another parameter $v$ of the Gaussian curve is also studied in Fig. 2l.

Interestingly, the result indicates that $b/B$ is identically 0.5 under all experimental conditions. Therefore, the geometric profiles of the HMP structures can be rewritten by

$$y = \alpha \left( \frac{B}{2} \tan\theta + t \right) e^{-\frac{(16\ln 2)x^2}{B^2}} \qquad (3)$$

where the two parameters $\alpha$, $t$ can be determined by the linear relationships with $B$ and $V$ according to the above results (Fig. 2i–k).

## Compressive behavior of the HMP

The compressive deformation of the micro-pyramid structure with different height ratios has been investigated to elucidate the mechanism of the hollow microstructures on high sensitivity over a wide range by finite element analysis (FEA). Figure 3a illustrates the result of the HMP, which serves as the structural dielectric layer of a CPS, being compressed by a plate. The deformation process can be categorized into three stages, as depicted in Fig. 3b. Under slight pressure, local stress concentrates at the contact point between the micro-pyramid and the top film, producing large deformation of the tip of the structure. As the increment of applied pressure, the cavity within the hollow microstructure experiences the majority of compressive deformation until it is eliminated under the exerted pressure (about 40 kPa). Subsequently, the structure primarily undergoes deformation via self-mutual extrusion, which resembles the behavior observed in solid microstructures. For the solid micro-pyramid ($h/H = 0$), stress consistently concentrates around the pyramidal tip under external pressure and is significantly larger than that of the HMP, which demonstrates higher resistance to external pressure (see more comparisons in Supplementary Fig. 10).

To further explore the deformation of the HMP, the compressive strains derived from the solid portion ($\varepsilon_{solid}$) and hollow portion ($\varepsilon_{hollow}$) are plotted in Fig. 3c, which elaborates on the three stages mentioned above. The compressive strain within the solid portion of the HMP (the blue line) quickly escalates under small pressures below 0.6 kPa, exhibiting a variation that is almost consistent with that of the solid structure (the red line). This can be attributed to the low compressive stiffness ($k$) at the pyramid's apex, which accounts for the high sensitivity in prior research[30,53]. However, with the pressure increasing, the growth of $\varepsilon_{solid}$ has substantially decelerated, signifying a significant descent in sensitivity. For HMPs, the compressive strain within the hollow portion (the blue dashed line) experiences a consistent and swift increase before the cavity is eliminated in stage 3, which can effectively compensate for that deficiency. Figure 3d displays the variations between the applied pressure and the total compression strain among the pyramid dielectric films with different height ratios. For the solid micro-pyramid, the pressure increases rapidly with the total compression strain, indicating a higher equivalent compressive stiffness, especially under large strains. Upon introducing an interior cavity into the micro-pyramid, the pressure is significantly reduced at the same compression strain. This indicates that as the pressure increases, the structural compressive stiffness rises more gradually, highlighting the potential of the proposed HMPs for use in CPSs to achieve high sensitivity over a wide range. Moreover, the relationships between the equivalent compressive stiffness $k$ and four key parameters of the micro-pyramid, including dimension parameters $h/H$, $B$, $t$, and Young's modulus $E$, are investigated in Supplementary Fig. 11. It is noted that the top width is much less than the base width according to the SEM image in Supplementary Fig. 12, illustrating that the actual shape of the micro-pyramid structure is typically represented as a frustum. A scaling law of the equivalent compressive stiffness can be derived as $k \propto Em(0.5\tan\theta + t/B)(1 + h/H)\ln(h/H)$ (see details in Fig. 3e and Supplementary Note 1), where $m$ denotes the top width of the pyramid structure. Distinctly, these two parameters of $h/H$ and $E$ contribute greatly to improving structural compressibility.

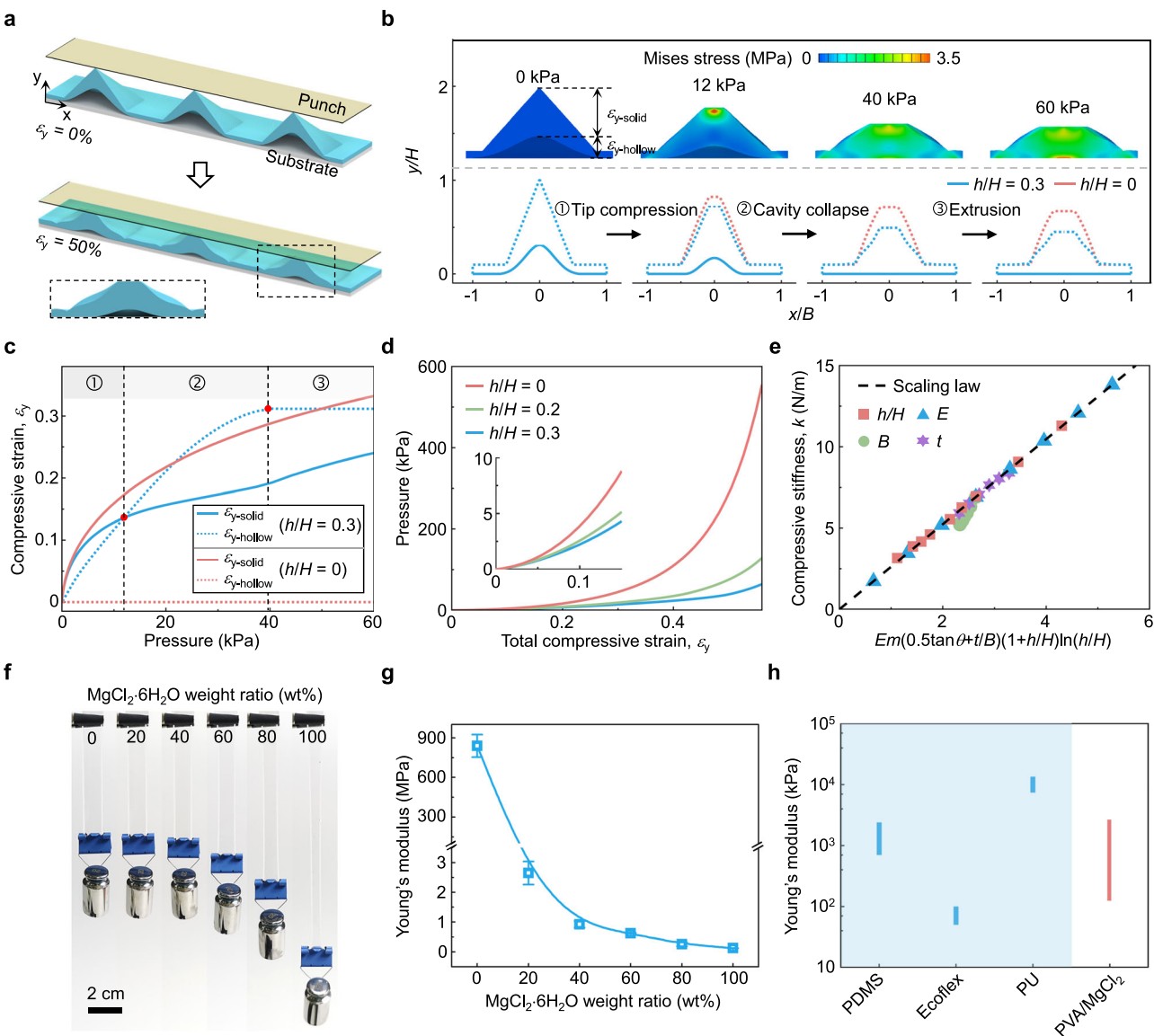

**Fig. 3 | Compressive behavior of the HMP structure. a** Illustration of the HMP being compressed by a plate. **b** FEA results of compressive deformation of the micro-pyramid structures. **c** Comparison of compressive strains derived from the solid portion ($\varepsilon_{solid}$) and hollow portion ($\varepsilon_{hollow}$) of solid and hollow pyramids. **d** Applied pressure changing with the total compressive strain under different height ratios. **e** Scaling law of the equivalent compressive stiffness as a function of the combined parameter $Em(0.5\tan\theta + t/B)(1 + h/H)\ln(h/H)$. **f** Optical images of the modified films with different weight ratios of $MgCl_2 \cdot 6H_2O$ to PVA from 0 to 100%, which are hung with a 20 g weight. **g** Young's modulus of the modified films versus the weight ratios of $MgCl_2 \cdot 6H_2O$. Error bars show s.d., $n = 3$. **h** Comparison of Young's modulus of $PVA/MgCl_2$ films with those of common elastomers.

The Young's modulus of the dielectric film can be tailored by adjusting the weight ratio of $MgCl_2 \cdot 6H_2O$ to PVA, ascribing to the disruption of hydrogen bonds and the decrease in intermolecular interactions between PVA chains (Supplementary Fig. 13)[54,55]. The results are presented in Fig. 3f, g. The pure PVA film exhibits poor softness with a much higher Young's modulus of 839 MPa (Supplementary Fig. 14), approximately 420 times higher than that of PDMS elastomer, which is widely used in flexible pressure sensors. Magnesium chloride has been demonstrated to possess a high plasticizing efficiency for PVA and can disperse within PVA at a molecular level, with excellent compatibility[55]. Upon doping the inorganic salt $MgCl_2 \cdot 6H_2O$ into PVA with weight ratios ranging from 20 to 100%, the modulus significantly decreases, spanning from 2.6 to 0.12 MPa, which are comparable to those of the PDMS and Ecoflex elastomers, respectively. As the doping ratio of $MgCl_2$ increases, the polymer film becomes softer, but elastic recovery also begins to deteriorate (Supplementary Fig. 15). At a doping ratio of 20%, the modified PVA film

demonstrates favorable rubber-like mechanical properties and an impressive elastic recovery rate of 99% under a tensile strain of 50%, which is selected to serve as the dielectric layer and fabricate the capacitive pressure sensor. With a further increase to 40% doping ratio, Young's modulus declines to 0.92 MPa, while maintaining a good elastic recovery ratio of 96%. Consequently, the proposed hollow microstructured dielectric film exhibits great potential as a viable alternative to commonly used elastomeric competitors (Fig. 3h) in developing high-performance CPSs[56,57]. In summary, through material and structure engineering, the hollow microstructured film becomes softer for higher sensing sensitivities and enables retarded structural stiffening during compression, which is favorable for an extensive pressure range.

## Performance evaluation of the HMP-enhanced CPS

A square pressure sensor with a dimension of $1\,cm \times 1\,cm$ is prepared to evaluate the sensing performance, as depicted in Fig. 4. To improve the

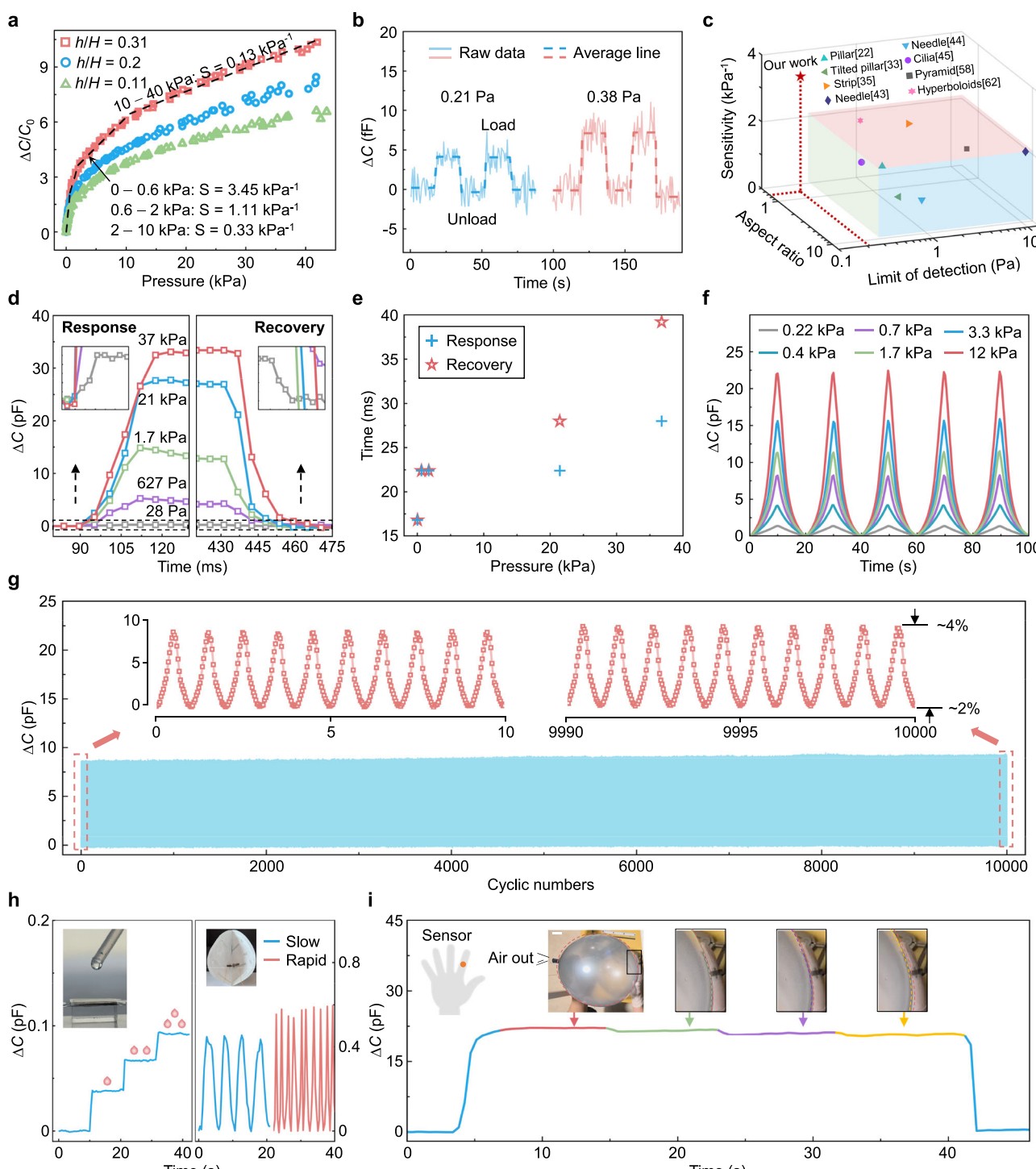

**Fig. 4 | Performances evaluation of the HMP-enhanced CPS. a** Pressure response of relative capacitance change of the CPS with different height ratios. **b** Detection of tiny pressures. **c** Comparison of the HMP-based CPS with previously reported microstructure-based CPSs in three key performance parameters: Sensitivity (preferred to be higher), limit of detection (preferred to be lower), and aspect ratio. **d**, **e** Response and recovery times under a range of applied pressures, from low (28 Pa) to high (37 kPa). **f** Dynamic pressure tests with different magnitudes of 0.22, 0.4, 0.7, 1.7, 3.3, and 12 kPa. **g** Reliability testing of a sample by cyclic loading with the amplitude of 1.2 kPa for 10,000 cycles. The insets present the capacitance changes at the first and last ten cycles. **h** Detection of three successive waterdrops and monitoring of human respiration status. **i** Sensing the tiny deflations of a balloon under a large preload of -12 kPa when the sensor is attached to the index finger.

homogeneity of compression load, a thin glass plate (1 cm × 1 cm, 27.5 mg) is placed on the capacitor surface. The sensitivity of a CPS is defined as $S = \delta(\Delta C/C_0)/\delta P$, where $C_0$ is the initial capacitance and $\Delta C$ is the capacitance change under the applied pressure ($P$), and is closely related to the compressive deformation of the intermediate dielectric layer. Compared to solid microstructured dielectric layers, hollow microstructured dielectric layers produce a greater compressive displacement (i.e., larger gap changes between capacitor electrodes) under the same pressure, yielding a larger relative capacitance change ($\Delta C/C_0$) for high sensitivity. The relative capacitance changes of the HMP-

enhanced CPS, prepared with different hollow-to-total height ratios of 0.11, 0.2, and 0.31 are shown in Fig. 4a, where the base width of the pyramid is 79 µm. With a height ratio of 0.31, the pressure sensor reaches a remarkable sensitivity of 3.45 kPa$^{-1}$ within the pressure range of 0−0.6 kPa. The sensitivity drops slightly with increasing pressure, yet remains at high levels of 1.11 kPa$^{-1}$ (0.6−2 kPa), 0.33 kPa$^{-1}$ (2−10 kPa), and 0.13 kPa$^{-1}$ (10−40 kPa) (the curves of absolute capacitance versus pressure plotted in Supplementary Fig. 16). The results demonstrate that a larger hollow height ratio, corresponding to a more significant interior cavity within the micro-pyramid, leads to higher sensing sensitivity. Such a general mechanical design by introducing an interior cavity is also expected to be effective in enhancing structural compressibility for various shapes of 3D microstructures. For instance, the sensing performances of the CPSs employing pyramid and dome structures with varying height ratios are compared, and a larger height ratio typically leads to an increased sensitivity (Supplementary Fig. 17). While maintaining the same structural parameters (including base width $B$, spacing $d$, and overall height $H$) and material parameter (Young's modulus $E$), the sensitivity of the HMP-enhanced CPS can be raised more than ten times compared to the solid microstructure-based CPS, which have been exemplified by the pyramid and dome. This improvement is also verified by the sensitivity comparison with prior publications based on micro-pyramid structure[53,58–61], as shown in Supplementary Table 1. Moreover, owing to the higher height-to-width aspect ratio of the pyramid (0.7) compared to the dome (0.5), the hollow pyramid-based sensor exhibits a distinct advantage in high sensitivity.

Additionally, the sensing performances of the HMP-enhanced CPS, relative to different material and geometric parameters, have been elaborated in detail (Supplementary Note 2). Young's modulus of the structured dielectric film, as another critical feature that affects the compressibility of the pressure sensor, has also been investigated to improve the sensing performance. The undoped PVA film with high elastic modulus exhibits large compressive stiffness, even engineered with HMP surfaces, and yields a low-pressure sensitivity comparable to that of the pressure sensor using a PDMS dielectric film with solid micro-pyramid surfaces. Upon doping MgCl$_2$ into PVA, the modulus notably decreases to 2.6 MPa, resulting in a substantial enhancement in sensitivity (Supplementary Fig. 18). A sparser distribution by spreading out the microstructures to increase the air-to-elastomer ratio, due to the ineffectiveness of air in resisting deformation, is favorable for higher sensitivity, which represents a simple and universal strategy applicable to all microstructure-based CPSs (Supplementary Figs. 19a–c, 20a). A smaller pyramid will be more favorable to obtain a higher sensitivity, which has been verified in other works (Supplementary Table 1). However, the interior cavity within a smaller micropyramid can be easily collapsed under pressure, limiting the high sensitivity to a small pressure range (Supplementary Fig. 20b). Figure 4b shows that the HMP-enhanced CPS can reliably detect the placement or removal of a tiny pinch of cotton (about 2.1 mg). A distinguishable stepwise increase in capacitance can be observed, corresponding to an average pressure of 0.21 Pa. After removing the cotton, the capacitance immediately returns to its initial value, and this variation recurs with subsequent loads. The tiny pressure change is also recognized when the pressure experiences a slight increase to 0.38 Pa. The detection limit of the developed sensor is in accordance with its sensitivity, which can be further optimized as discussed in Supplementary Note 2. Furthermore, the step loading-unloading pressure test is performed to describe the sensor behavior under static loads from 33 to 121 Pa (Supplementary Fig. 21a). The developed sensor based on a hollow micro-pyramid exhibits good stability under different levels of external loads, where the capacitance signals remain almost horizontal during each step pressure and change without hysteresis. Compared to reported structural designs of dielectric layers that aim to improve aspect ratios through developing advanced, complicated techniques, including various pillar or needle microstructures, hollow

microstructures denote another facile and effective direction for achieving high sensitivity. Although the pyramidal structure has a low aspect ratio of 0.7, the HMP-enhanced CPS demonstrates significant advantages in terms of sensitivity and limit of detection[22,33,35,43–45,58,62], as shown in Fig. 4c. Moreover, due to the merits of hollow microstructures, such as exceptional compressibility and superior resistance to structural stiffening, this mechanical design is promising for using in other types of pressure sensors (e.g., piezoresistive, iontronic, and triboelectric sensing mechanisms) for high performance.

In addition, the response performance of the HMP-enhanced CPS to dynamic loads has been investigated as well. The response and recovery times under various loads, ranging from 28 Pa to 37 kPa, are determined (Fig. 4d and Supplementary Fig. 21b). The device is subjected to external pressure, which is then rapidly released after being sustained for a period. As the pressure increases, the response and recovery times become longer, which can be explained by the greater compression displacements. In Fig. 4e, the shortest response and recovery times are both 16.8 ms at $P = 28$ Pa, and the longest response and recovery times are 28 ms and 39.2 ms at $P = 37$ kPa, respectively. The fast response speed of developed CPS allows it to detect dynamic stimuli, such as robotic tactile sensing[10,63], and artery pulse detection[64]. The HMP-enhanced CPS is able to reliably detect varying levels of dynamic pressure in a highly repeatable way (Fig. 4f). To evaluate the reversibility and durability of the device, the response performance is investigated by repeatedly loading/unloading a pressure pulse (1.2 kPa) in Fig. 4g. The signal exhibits minimal changes after 5000 and 10,000 cycles, i.e., the minimum capacitance increased by 1.46 and 2.21%, and the peak capacitance increased by 2.58 and 4.03%, respectively.

Since environmental temperature and relative humidity commonly affect the response of capacitive devices, the stability of developed sensors under varying temperatures and humidity is also essential to practical applications. Here, the developed pressure sensor is well packaged, such that the humidity change does not affect the signal (Supplementary Fig. 22a). Supplementary Fig. 22b presents the response of the developed sensor in various ambient temperatures from 25 to 45 °C, which covers the temperature of the human body. The difference in relative capacitance variation of the developed sensor is a little under the same load (5 g).

Figure 4h, i present the applications of HMP-enhanced sensors in various practical scenarios, such as tiny pressures encountered by human skin on a regular basis or large pressures by grasping. In the left of Fig. 4h, this sensor without preload could differentiate the static pressures induced by three waterdrops applied one after another, corresponding to approximately 24.2 mg for a drop of water. When the developed sensor is integrated with the inner surface of a mask, it can be used for monitoring the respiratory status of the human to release early-warning signals, e.g., slow and rapid respiration of ~12 times per minute (blue curves) and 32 times per minute (red curves) in the right of Fig. 4h, respectively. Additionally, the HMP-enhanced pressure sensor can also be attached to the index finger for tactile sensing with approximately 12 kPa preload (e.g., grasping the balloon). The initial shape of the balloon is marked with red dotted lines in the inset of Fig. 4i. Then, three consecutive tiny deflations of the balloon produce three consecutive stepwise decreases in pressure, which is hard to feel by hand, but can be detected by the developed sensor. The zoom-in insets illustrate the subtle shape changes of the balloon under three tiny deflations, which are marked with green, purple, and yellow dotted lines, respectively. These experiments demonstrate that the HMP-enhanced pressure sensor has a high sensitivity and robustness over a wide pressure range.

## Demonstration of the HMP-enhanced CPS
Encouraged by the impressive performances of HMP-enhanced CPS, the application of developed sensors in the field of robotic tactile is

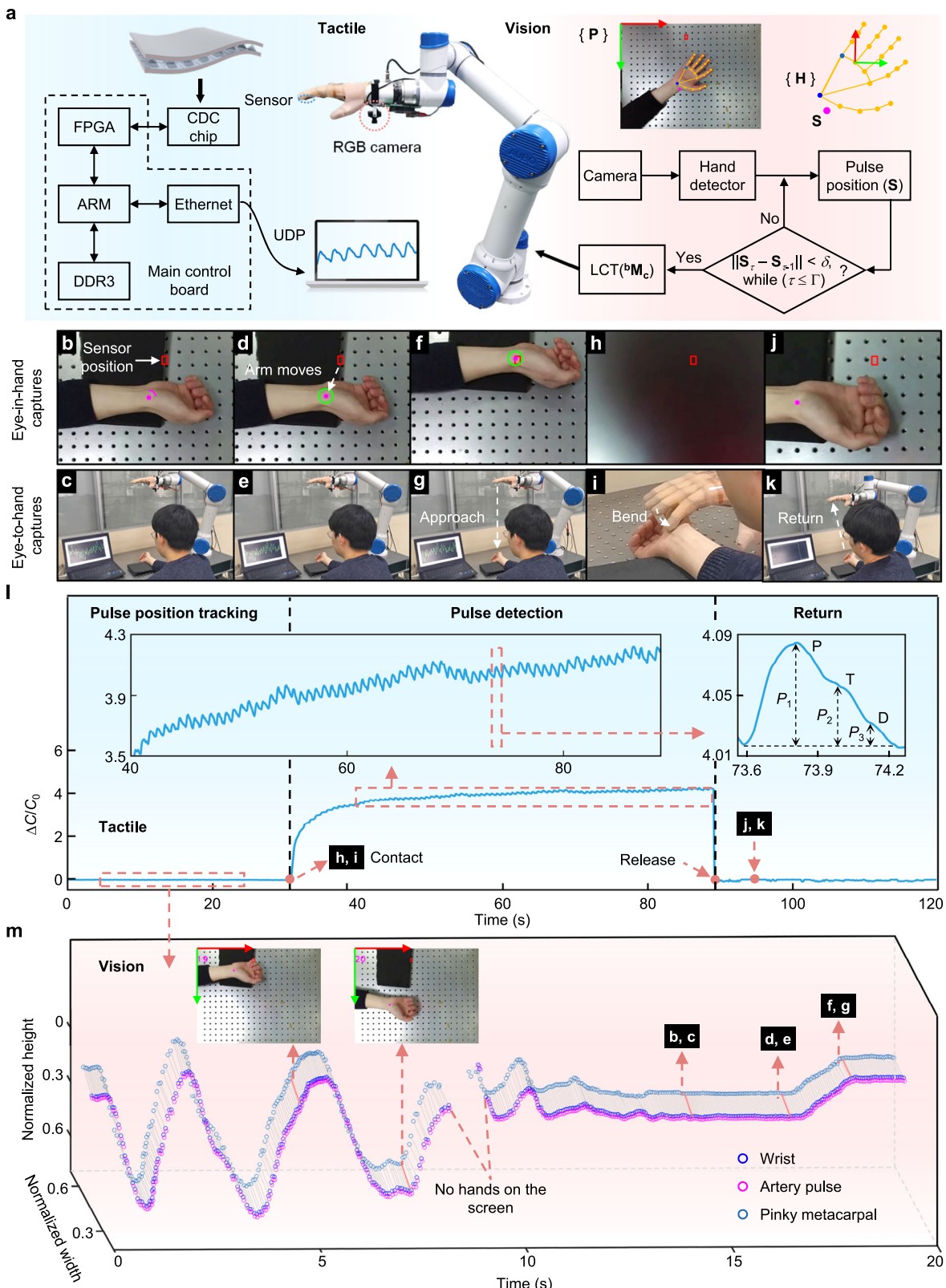

explored, which always involves detecting small pressures under large pre-contact pressure. To demonstrate this concept, the autonomous pulse-diagnosis robotic system (APDRS) is developed that enables a robot to detect pulse signals like a doctor. The system comprises three components: tactile sensing based on the HMP-enhanced CPS, vision-assisted location of pulse-diagnosis point, and a robot arm outfitted with a bionic hand at the end, as depicted in Fig. 5a. The

HMP-enhanced CPS is attached to the index fingertip of the bionic robot hand to mimic the human hand feeling pulse beats. The self-developed circuits acquire the sensor signals with a sampling rate of 200 Hz, which is high enough for pulse detection. The capacitance values are firstly converted into stable digital signals by a capacitance-to-digital converter (CDC) chip. The digital signals are then transmitted to the FPGA (Field-programmable gate array) main control

**Fig. 5 | Demonstration of the HMP-enhanced CPS applied in automatic pulse-diagnosis robotic system (APDRS). a** Schematic illustration of the APDRS construction, including the HMP-enhanced CPS-enabled tactile sensing, self-developed capacitance acquisition circuits, vision-aided localization algorithm for determining the pulse-diagnosis point, and a robot arm outfitted with a dexterous hand at the end. Advanced RISC Machine (ARM) is a family of reduced instruction set computing (RISC) architectures designed for computer processors, Double data rate type 3 (DDR3) is a type of synchronous dynamic random access memory used for system memory, and user datagram protocol (UDP) is transport layer protocol. The workflow is illustrated by **b**–**j** eye-in-hand captures and **c**–**k** eye-to-hand captures. **b**, **c** In the first step, the pulse-diagnosis point is recognized using the machine vision technique and denoted with a magenta dot. **d**, **e** When the human hand remains stationary for a while and the circular progress bar turns green, the robot arm moves horizontally to allow the sensor (marked by a red rectangle) on the index fingertip of the dexterous hand to overlap with the pulse point and (**f**, **g**) then approaches downward. **h**, **i** The index finger of the robot hand bends to touch the human hand for pulse detection. **j**, **k** Upon completion, the robot arm returns to the initial position for the next patient. **l**, **m** Evolution of **l** the HMP-enhanced CPS signals and **m** the wrist/artery pulse/pinky metacarpal points throughout the detection process.

board and ultimately sent to the host computer (data collected by the commercial package NI Labview 2017) via Ethernet communication. By employing a monocular camera mounted on the robotic hand, the location information of the pulse-diagnosis point (**S**) can be accurately predicted, utilizing a combination of the coordinate transformation of monocular imaging and an image processing technique (see details in Supplementary Fig. 23 and Note 3).

Figure 5b–k demonstrate the workflow of the APDRS, while Fig. 5l, m present the evolution of the electrical signals and vision data during the detection process, respectively. When a human hand appears in the capture, the position of the pulse-diagnosis point is quickly determined and marked with a magenta dot in Fig. 5b. To verify the system's robustness, we wave the hand casually in the captures, the system can still track the pulse-diagnosis point instantly and precisely with a stable frame rate of 20, while the real-time position signals also display a similar waveform in Fig. 5m. If there are no hands in the captures, a corresponding gap of position signal will appear. If the human hand moves onto the pillow and remains stationary for a preset time ($\Gamma = 5$ s, corresponding to the condition of $||\mathbf{S}_\tau - \mathbf{S}_{\tau-1}|| < \delta$ while $\tau \le \Gamma$, where $\delta$ is a distance threshold), the position signal will remain unchanged, indicating that the patient is ready for pulse detection. Following the linear coordinate transformation (LCT, $^b\mathbf{M_c}$), the robot arm moves to the target position and approaches the human arm (Fig. 5f, g). Afterward, the robot index finger equipped with the HMP-enhanced CPS bends to contact with the human arm (Fig. 5h, i). Upon contacting the pulse-diagnosis point, the capacitance signals immediately rise and the typical pulse waveforms (e.g., percussion, tidal, and diastolic peaks) appear, as shown in Fig. 5l. Because the fingers of the robot are relatively hard and there is a noticeable amount of contact pressure, it will result in slight discomfort and wobble of volunteer's wrist during the pulse detection, leading to the baseline slight fluctuations of the capacitive signal. This will take time for a volunteer to adapt to the press from a robotic hand. It is obversed that the baseline of the signal becomes stable as the longer measurement time. The pulse signals can be consistently reproduced with an average frequency of ≈82 beats per minute, in accordance with the normal adult values. Based on these three peaks, the radial augmentation index ($AI_r = P_2/P_1$), diastolic augmentation index ($DAI_r = P_3/P_1$), and digital volume pulse time ($\Delta T_{DVP} = t_{P2} - t_{P1}$) under relaxed conditions are calculated to be 0.578, 0.222, and 0.402, respectively, which can be used to diagnose arterial stiffness[21,65]. Finally, after the robot finger releases and returns to the initial position (Fig. 5j, k), the capacitance value of the developed CPS quickly returns to the initial level as well. While the pressure sensor experiences a large pre-contact pressure (about 4 kPa), the small pulse pressure changes can still be recognized clearly and continuously throughout the detection process, which lasts for about one minute. This is attributed to enough sensitivity and excellent stability of HMP-enhanced CPS over a large dynamic range, promising in robotic tactile.

Additionally, for localized pressure detection, a 3 × 3 sensing array with HMP-enhanced capacitive sensors has been constructed (Supplementary Fig. 24). The response of the sensing array is precisely reflected in corresponding pressure mapping under a cylinder for single input and an H-shaped acrylic plate for multiple inputs, demonstrating a little influence between the sensor units within the array. For a micro-pressure array, a spacer structure can be inserted between the top and bottom electrode layers of the sensing array to create a strain local confinement effect and effectively eliminate the spatial crosstalk between sensing units under external pressure[14,24].

## Discussion

We propose a Marangoni-driven deterministic formation approach for fabricating hollow microstructures, which display the distinct advantages of being softer and possessing more room for compression. A combined strategy of altering solution properties and enhancing evaporation-induced convective deposition is utilized to precisely control the interior cavity within micro-pyramids, achieving a high void ratio exceeding 90%. Concurrently, a geometric model is established to accurately describe the cavity profile with curve-fitting RMSEs of below 2 μm. When served as the dielectric layer of a non-intronic CPS, the results indicate that the HMP-enhanced CPS exhibits an ultra-low LoD of 0.21 Pa, and a tenfold increase in sensitivity over the pressure sensor utilizing solid micro-pyramids of identical dimensions. Compared to the direction of improving the aspect ratio to create needle-shaped microstructures through developing advanced, complicated techniques, the HMP structure has a low aspect ratio of 0.7, yet delivers significant advantages in high sensitivities over a wide pressure range. The unique benefits stem from the collapse of the internal cavity, which permits more substantial deformation, significantly boosting compressibility while retarding structural stiffening during compression. The ARPDS has been constructed to demonstrate the great potential of the HMP-enhanced CPS in robot tactile, which often involves detecting subtle pressure changes under a large pre-contact pressure, such as arterial pulse detection.

We believe that such a facile, scalable, and large-area compatible technique for fabricating hollow microstructures in various geometric shapes, as an alternative to current solid microstructures, provides a general strategy for addressing the longstanding challenge of high sensitivity over a broad pressure range and will be extended to other types of pressure sensor (e.g., piezoresistive, iontronic, and triboelectric sensing mechanisms).

## Methods

### Preparation of the PVA/MgCl₂ solution
Firstly, 100 g of deionized water is mixed with 2 g of polyvinyl alcohol (PVA 1788 from Usolf Corporation), which is stirred at 95 °C for 2 h until PVA completely dissolves. Then, $MgCl_2 \cdot 6H_2O$ (Sinopharm Chemical Reagent Co., Ltd) is added into the PVA solution and stirred at 60 °C for 2 h to obtain the PVA/MgCl₂ solution. Different weight ratios of $MgCl_2 \cdot 6H_2O$ to PVA of 0, 20, 40, 60, 80, and 100% are produced for comparisons. Finally, a weight ratio of 20% is employed to fabricate the HMP-enhanced CPS.

### Preparation of PDMS/AgNWs/PVA electrode layer
The pure PVA solution is prepared by dissolving 2 g of polyvinyl alcohol into 25 g of deionized water and stirring at 95 °C for 2 h. Then, the PVA film is obtained by pouring the PVA solution onto a 75 mm × 25 mm glass slide and curing at room temperature for 24 h.

A self-adhesive paper is patterned by a cutting machine (GRAPHTEC Corporation, Cutting Plotter CE6000-40, Japan) and attached to the PVA film as a shadow mask. The AgNWs solution dissolving in isopropyl alcohol is sprayed on the PVA film and dried at 50 °C for 0.5 h to produce the electrode with a size of 10 mm × 10 mm. Specifically, for the bottom electrode layer, the prepared PDMS (Sylgard 184, Dow Corning) with a weight ratio of base-to-crosslinker of 10:1 is spread on the PVA film, followed by spin coating at 3000 rpm for 40 s, and cured at 70 °C for 8 h, which is acted as a thin insulating layer (<10 μm). This PDMS insulating layer prevents ionic conduction between the AgNWs electrode and PVA/MgCl$_2$ film and enhances the sensor reproducibility[61]. Differently, for the top electrode layer, the spin-coating speed is 300 rpm to obtain a thick PDMS layer (-230 μm) that serves as an encapsulation layer of the sensor, providing environmental isolation and physical protection.

### Fabrication of the HMP-enhanced pressure sensor
The fabrication process is shown in Supplementary Fig. 25. As a result of the strong adhesion between the silicon mold and the polymer film, the polymer film experiences elongation and fractures instead of interfacial peeling during the peel-off test, with a maximum force of up to 1.2 N (Supplementary Fig. 26). Conversely, the maximum force for successful peeling from the PDMS mold is about 0.3 N. Therefore, two-step replication processes are performed to obtain the PDMS mold with inverted micro-pyramid structures. Firstly, the PDMS mixture is poured onto the silicon mold with etched pyramid micropatterns to obtain an inverse structured template. After fluorination treatment to weaken adhesion, the molded pattern is replicated again to produce the second PDMS mold. In order to prevent the deformation of the molding template in use, hard PDMS (with a base-to-crosslinker ratio of 5:1) is employed in the second molding process, which has a larger Young's modulus than standard PDMS (with a base-to-crosslinker ratio of 10:1)[66]. The cross-sectional SEM images of the first and second PDMS molding templates are shown in Supplementary Fig. 27a, b, respectively. Four silicon molds with different base widths of 39, 79, 125, and 141 μm are tried in this work. The pyramid array is replicated precisely without changes in dimensions and side angles (54.7°) (Supplementary Fig. 27c, d). Next, the prepared PVA/MgCl$_2$ solution is poured onto the second PDMS mold to form the HMP film. Finally, the bottom and top electrode layers, consisting of bottom-up stacked PDMS, AgNWs, and PVA layers, are laminated with the HMP film to obtain the pressure sensor.

### Measurements of mechanical and electrical properties
Three independent polymer films are prepared by spreading the PVA/MgCl$_2$ solution on a glass sheet and curing it at room temperature for 24 h. The polymer films are cut into the size of 70 mm × 15 mm. The thickness of films is determined by averaging measuring results of three different parts of the sample using laser scanning confocal microscopy (LSCM, KEYENCE, VK-X200K). The samples are tested by the tensile tester (INSTRON 5944) to obtain Young's modulus of the polymer films. In addition, the capacitance of the self-developed sensor is acquired by the LCR meter (E4980A, KEYSIGHT) at a testing frequency of 1 kHz. The applied pressures are recorded by using a force gauge (DS2-5N, Zhiqu Precision Instruments Co., Ltd) with a resolution of 1 mN, as shown in Supplementary Fig. 28.

### Compressive simulation by finite element analysis
Finite element models of micro-pyramid structures are constructed by using commercial software ABAQUS 6.14-1. The compressive deformation of PVA/MgCl$_2$ microstructured film is simulated with the Mooney-Rivlin hyperelastic model, where the material parameters are determined by the standard uniaxial tensile test (see details in Supplementary Fig. 29 and Table 2). Eight-node solid elements (C3D8H), and analytic rigid are used to model the micro-pyramid structures and

indenter, respectively. The adoption of refined meshes ensures the precision of numerical simulations. To avoid mesh distortion heavily, the top of the pyramid is treated with a very small width (1 μm).

### Peel tests
The interface bond strength between the microstructured film and the PDMS/Si molds is measured by the self-developed platform (Supplementary Fig. 26). The mold is fixed on the baseplate with a tilt angle of 45°. The microstructured film formed on the mold is connected to a force gauge via a thick Kapton adhesive tape (-200 μm). The peeling speed in the vertical direction is set to 0.1 mm/s.

### Arterial pulse detection
The experiment of detecting arterial pulse is conducted with the approval of the Ethics Committee of Tongji Medical College, Huazhong University of Science and Technology. This Ethics Committee is constituted and functioned in accordance with ICH-GCP, GCP in China, and the Declaration of Helsinki (2013). Informed consent is obtained from the participants before conducting the experiments. The 41 images of human hands with different gestures and/or arm sizes, which are used to train the vision-aided location prediction model in Supplementary Fig. 23, are collected from the volunteers (aged 20–50 years) of our study group. The images, without any labels, are randomly allocated into the training and testing groups. The arterial pulse data (in Fig. 5) are captured from a healthy male, aged 25–30 years. There is no specific preparation for the human subject. These experimental results are not specific to any particular sex or gender without incorporating sex and gender into the study design.

### Statistics and reproducibility
The SEM images of the hollow micro-pyramid structures are collected from more than three independent samples. Arterial pulse detections are conducted more than three times and all testing results show high similarity.

### Reporting summary
Further information on research design is available in the Nature Portfolio Reporting Summary linked to this article.

## Data availability
The data that support the findings of this study are available within this article and its Supplementary Information, and from the corresponding authors upon request.

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

## Acknowledgements
This work was supported by the National Natural Science Foundation of China (Grant 51925503 to Y.H., Grant 52375568 to F.Z., and Grant 52188102 to Y.H.), the XPLORER Prize (Grant 2020-1036 to Y.H.), and the Science and Technology Innovation Team of Hubei Province. The authors thank the Flexible Electronics Manufacturing Laboratory in the Comprehensive Experiment Center at Huazhong University of Science and Technology for support in advanced manufacturing equipment.

## Author contributions
Y.H., F.Z., and W.X. conceived the idea and led research efforts. W.X. performed the experiments. S.Q. helped with the preparation of HMP films and the sensor testing. Y.H., F.Z., and K.L. participated in the discussion of experimental results. W.X. wrote the manuscript and designed the figures. F.Z. and L.Y. participated in the design of the figures. Y.H. and F.Z. revised the manuscript and supervised the research. All authors reviewed and commented on the paper.

## Competing interests
The authors declare no competing interests.
