## [Peer Review File · Nature Communications]

Marangoni-driven Deterministic Formation of Softer, Hollow Microstructures for Sensitivity-enhanced Tactile SystemREVIEWER COMMENTS

Reviewer #1 (Remarks to the Author):

This article presents the design of a Marangoni-driven hollow pyramid-shaped microstructure flexible sensor, which exhibits ultra-light pressure detection and superior capacitive sensing capabilities compared to solid microstructures. By controlling parameters such as evaporation temperature and the content of $MgCl_2$, different microstructures with varying h/H ratios and Young's moduli can be obtained and reflected in the fitted formula. Therefore, I recommend accepting this article with major revision.

1. What led to the selection of $MgCl_2$ as the additive for modifying the Young's modulus? What are the advantages?
2. Can the thin film in Figure 3f recover its shape after being stretched?
3. In the article, it is mentioned that we can accurately predict the pulse location and react to hand movements accordingly. However, is our prediction model applicable to everyone? It would be beneficial to conduct tests with additional volunteers and provide the accuracy of the predictions.

Reviewer #2 (Remarks to the Author):

In this paper, the authors present a Marangoni-driven deterministic formation approach for fabricating hollow structured microarrays while exploring the influences of significant process parameters. The mechanical properties of hollow microstructures were evaluated through experiments and numerical simulation. The hollow microstructural film can serve as the dielectric layer to significantly enhance the sensitivity and detect limit of capacitive tactile sensors, which has wide potential for practical applications, such as robot tactile. The proposed approach for producing hollow microstructures is scalable and suitable for flexible, large-area microstructures and sensors manufacturing, which is a significant contribution to the field. I think it is more than appropriate for publication in Nature Communication after minor revision.

Comments:

- 1.The microstructures are widely utilized in Flexible tactile sensors, which can be produced in various methods. In terms of applications, the authors should provide better discussion and comparison of the methods for creating hollow microstructures and Marangoni-driven formation approaches.
- 2.The performance of the prepared sensor is significantly impacted by the shape of microstructures. The Figure 1b-d shows the shapes of hollow pyramid microstructures. Whether the shapes of these structures are consistent. Can the approach be applied to forming different hollow microstructures?
- 3.The formation of the sensor is important for practical applications. What are the roles of the different layers of hollow micro-pyramid enhanced CPS sensors, such as the PVA layer and PDMS layer in Figure 1f?
- 4.The theoretical model would be helpful for the design of the tactile sensor. The scaling law is good. But more details of scaling law of the equivalent compressive stiffness should be included in this paper, particularly about the definition of "m" in Figure 3e.
- 5.In Figure 4b, the authors mention that the low limit of detection of the prepared sensor is 0.21 Pa. Can it be decreased further in the future?
- 6.There is an inconsistency between the references in Figure 4c and the references in the main body of the manuscript.
- 7.In Figure 5a, the definition of ARM, DDR3, and UDP needs to be explained clearly.

Reviewer #3 (Remarks to the Author):

The manuscript titled "Marangoni-driven deterministic formation of softer, hollow microstructures for sensitivity-enhanced tactile system" reported a wearable capacitive tactile sensor based on a hollow micro-pyramid microstructure. The controllable drying process of the polymer solution

rendered adjustable interior cavity properties, high sensitivity, and an extensive measurement range. The authors tried to present a comprehensive understanding of the working mechanism, encompassing fundamental sciences and practical applications. Generally, this manuscript exhibits good organization, logical clarity, and a balanced integration of experiments and theories. However, specific concerns require attention, which are outlined below.

1. What standards define the base width range of hollow microstructures? The size and distribution of microstructures significantly impact sensitivity and detection range, and the author should furnish evidence supporting the optimization of the current selection. Additionally, if different sizes and distributions of microstructure combinations can achieve superior results?
2. The authors are recommended to elaborate on achieving precise detection and sensing of local pressure within a confined device area and address the potential influence of microarray electrode preparation on sensitivity and robustness.
3. The author could demonstrate more practical scenarios that span various pressure ranges to highlight the device's sensitivity and robustness advantages and provide practical applications to showcase the versatility of the proposed sensor.
4. Please investigate the baseline behavior over extended pulse signal testing periods (refer to Fig. 5I), explore the reasons for baseline monotonic fluctuations, and assess the impact of physical activity on pulse testing in portable wearable systems. Additionally, the authors are recommended to discuss strategies to reconcile device sensitivity with external environmental interference in practical applications.

Detailed response to referees' comments

Responses to comments of Referee #1

Summary Comment: “This article presents the design of a Marangoni-driven hollow pyramid-shaped microstructure flexible sensor, which exhibits ultra-light pressure detection and superior capacitive sensing capabilities compared to solid microstructures. By controlling parameters such as evaporation temperature and the content of MgCl_2 , different microstructures with varying h/H ratios and Young's moduli can be obtained and reflected in the fitted formula. Therefore, I recommend accepting this article with major revision.”

Reply: We thank the referee for the constructive comments that help to improve our manuscript considerably. We have carefully addressed all the issues detailed below and revised our manuscript accordingly.

Comment 1: “What led to the selection of MgCl_2 as the additive for modifying the Young's modulus? What are the advantages?”

Reply: We thank the referee for this comment. Young's modulus of the dielectric layer has a significant effect on the performance of capacitive sensors. Here, the magnesium chloride (MgCl_2) serves as a novel and highly efficient plasticizer for polyvinyl alcohol (PVA), enabling a PDMS-like property of the dielectric film, and thereby improving the sensitivity of developed capacitive sensor.

The pure PVA film is not sufficiently soft because the mobility of PVA chains is restricted by its intramolecular or intermolecular strong hydrogen bonds¹. Extensive efforts have been made to

weaken the hydrogen bonding in order to enhance the flexibility of PVA film. Previous research has proved that MgCl_2 exhibits high plasticizing efficiency for PVA and can disperse in PVA at a molecular level with good compatibility².

The compressive behavior of the hollow micro-pyramid in Fig. 3 and Supplementary Fig. S9 of the original supplementary information indicates that the equivalent compressive stiffness of the microstructured dielectric film (i.e., PVA film) increases linearly with its Young's modulus. The Young's modulus of the pure PVA film (Fig. 3g of the original manuscript) is 839 MPa, which is approximately 420 times higher than that of polydimethylsiloxane (PDMS) elastomer, widely used in flexible pressure sensors³. Therefore, when serving as the dielectric layer of a capacitive pressure sensor, the high modulus of the pure PVA film will still lead to an extremely low sensitivity even after the hollow structure is introduced into the micro-pyramid. Supplementary Fig. S10 of the original supplementary information illustrates that magnesium ions (Mg^{2+}) can disrupt the hydrogen bonds and form PVA- Mg^{2+} chelate by combining with the hydroxyl groups of PVA. This molecular structure weakens intermolecular forces and reduces the entanglements between PVA chains. According to the results in Fig. 3g of the original manuscript, the modulus of modified PVA film significantly decreases to 2.6 MPa after doping the inorganic salt MgCl_2 into PVA with a weight ratio of 20%, which is comparable to that of the PDMS elastomer.

References

1. Kubo, J. I., *et al.* Improvement of poly(vinyl alcohol) properties by the addition of magnesium nitrate. *J. Appl. Polym. Sci.* **112**, 1647-1652 (2009).

2. Jiang, X., Zhang, X., Ye, D., Zhang, X. & Dai, H. Modification of poly(vinyl alcohol) films by the addition of magnesium chloride hexahydrate. *Polym. Eng. Sci.* **52**, 1565-1570 (2012).
3. Vaicekauskaite, J., Mazurek, P., Vudayagiri, S. & Skov, A. L. Mapping the mechanical and electrical properties of commercial silicone elastomer formulations for stretchable transducers. *J. Mater. Chem. C* **8**, 1273-1279 (2020).

Modification to the manuscript:

On page 14, we changed “The pure PVA film exhibits poor softness with a much higher Young’s modulus of 839 MPa (Supplementary Fig. S14), approximately 420 times higher than that of PDMS elastomer, which is widely used in flexible pressure sensors. Magnesium chloride has been demonstrated to possess a high plasticizing efficiency for PVA and can disperse within PVA at a molecular level, with excellent compatibility⁵⁶.”

Comment 2: “Can the thin film in Figure 3f recover its shape after being stretched?”

Reply: We thank the referee for this comment. Here, the modified PVA film with a doping weight ratio of 20% is selected to serve as the dielectric layer and fabricate the developed capacitive pressure sensor, which can be completely recuperative after unloading.

The elastic recovery rate of PVA films with various doping ratios of $\text{MgCl}_2 \cdot 6\text{H}_2\text{O}$ has been investigated through additional tests. As shown in Fig. R1a, the films are cut into straight strips (75 mm \times 10 mm) and placed on a stretching platform, with fixing two ends of the strips. These strips are stretched by 50% and held for approximately 30 s before being released. The elastic recovery rate (ER) can be calculated through the formula $ER = (L_1 - L_2) / (L_1 - L_0)$, where L_0 is the initial length of the straight strip, L_1 is the length after being stretched by 50%, and L_2 is the recovery length after

release the external loads. The undoped PVA film demonstrates a high Young's modulus of 839 MPa and an elastic recovery rate of 96.5% (Fig. R1b). Upon doping $\text{MgCl}_2 \cdot 6\text{H}_2\text{O}$ into PVA with a weight ratio of 20%, the modified PVA film exhibits favorable rubber-like mechanical properties, with Young's modulus of 2.6 MPa and elastic recovery rate of 99% under a tensile strain of 50%. As the doping ratio of MgCl_2 increases, the polymer film becomes softer, but elastic recovery also begins to deteriorate. At a weight ratio of 100%, it becomes challenging for the polymer film to return to its initial length, where elastic recovery rate is only 38.7%.

Fig. R1. Elastic recovery rate of the modified PVA films with different doping ratios of $\text{MgCl}_2 \cdot 6\text{H}_2\text{O}$. (a) Schematic illustration and parameters (initial length L_0 , deformed length L_1 with a stretching strain of 50%, and recovery length L_2) of elastic recovery rate experiments. (b) Elastic recovery rate of the modified PVA film under different levels of $\text{MgCl}_2 \cdot 6\text{H}_2\text{O}$ weight ratio.

Modification to the manuscript:

i) On pages 14 and 15, we added “As the doping ratio of MgCl_2 increases, the polymer film becomes softer, but elastic recovery also begins to deteriorate (Supplementary Fig. S15). At a doping

ratio of 20%, the modified PVA film demonstrates favorable rubber-like mechanical properties and an impressive elastic recovery rate of 99% under a tensile strain of 50%, which is selected to serve as the dielectric layer and fabricate the capacitive pressure sensor. With a further increase to 40% doping ratio, the Young's modulus declines to 0.92 MPa , while maintaining a good elastic recovery ratio of 96%.”

ii) We also added **Fig. R1** as **Supplementary Fig. S15** in the revised supplementary information.

Comment 3: “In the article, it is mentioned that we can accurately predict the pulse location and react to hand movements accordingly. However, is our prediction model applicable to everyone? It would be beneficial to conduct tests with additional volunteers and provide the accuracy of the predictions.”

Reply: We thank the referee for pointing out this issue. We think that the vision-aided location prediction model is applicable to a variety of individuals, as shown in Fig. R2. To demonstrate the robustness and accuracy of this model, additional samples from various volunteers (e.g., with different gestures and/or arm sizes) have been gathered to evaluate the prediction model.

Different persons' hands are captured as shown in Fig. R2a for training the vision-aided location prediction model. The pulse-diagnosis point (i.e., highest beating) is determined by gently touching the volunteer's wrist with our fingertip, which is recorded as an observation value. Subsequently, the hand images are fed to the trained hand detector module to acquire the hand-knuckle points in the pixel coordinate system (PCS)¹, which are marked with solid green dots in Fig. R2a. To provide a clearer description of the artery pulse point's position in the image, a hand coordinate system (HCS) is

established, with the ring finger metacarpal (MCP) point serving as the origin. Hence, the transformation matrix (pM_h) from HCS to PCS can be expressed as,

$${}^pM_h = \begin{bmatrix} \cos \theta & -\sin \theta & x_{\text{ring_finger_mcp}}^p \\ \sin \theta & \cos \theta & y_{\text{ring_finger_mcp}}^p \\ 0 & 0 & 1 \end{bmatrix}, \quad (1)$$

and θ is the rotation angle between the coordinate systems and can be calculated as,

$$\theta = -\arccos \left(\frac{x_{\text{pinky_mcp}}^p - x_{\text{ring_finger_mcp}}^p}{\|S_{\text{pinky_mcp}}^p - S_{\text{ring_finger_mcp}}^p\|} \right), \quad (2)$$

where $S_{\text{pinky_mcp}}^p$ and $S_{\text{ring_finger_mcp}}^p$ are the coordinates of the pinky metacarpal and ring finger metacarpal points in PCS, respectively. Multi-output support vector regression (M-SVR) algorithm, capable of outputting multiple predicted values together, is utilized to predict the position of the pulse point (S) in HCS ². Four key points [i.e., the wrist, thumb carpometacarpal (CMC), index finger metacarpal, and middle finger metacarpal] serve as the inputs (X) of the M-SVR algorithm, which can be written as,

$$X = {}^pM_h^{-1} [S_{\text{wrist}}^p, S_{\text{thumb_cmc}}^p, S_{\text{index_mcp}}^p, S_{\text{middle_mcp}}^p]. \quad (3)$$

Next, the output (Y) is converted into the position in the PCS by the transformation matrix (pM_h) and indicated by a magenta circle in the image. A total of 33 samples are fed to train the vision-aided location prediction model based on the M-SVR algorithm. The fitting results are presented in Fig. R2a and b, where the coefficient of determination (R^2 score) is calculated as 0.984 (preferred to be close to 1), and the root mean square error (RMSE) and the mean absolute error (MAE) are relatively small and equal to 10.469 pixels and 7.739 pixels, respectively. Moreover, the ratio of RMSE to MAE, equal to 1.353, is close to $\sqrt{\pi/2}$ (≈ 1.253), suggesting that the fitting errors follow a normal distribution. From the training data in Fig. R2a, the predicted pulse point (marked with a magenta

circle) is closely consistent with the observed pulse point (marked with a blue circle), with specific coordinate values depicted in Fig. R2b. These results show that the M-SVR algorithm has a high goodness of fit.

To further validate the prediction accuracy of the model, hand images with different gestures and/or arm sizes from an additional eight independent volunteers are processed using the trained model described above. The results, shown at the bottom of Fig. R2a and Fig. R2c, indicate that the predicted pulse point closely accords with the observed pulse point. The RMSE and MAE are exceptionally small, as 6.314 pixels and 3.875 pixels, respectively. These results highlight the capability of the described workflow to accurately predict the pulse position based on hand captures from a monocular camera by using the vision-aided location prediction model.

Fig. R2. Schematic illustration and prediction results of the vision-assisted location of pulse-diagnosis point based on a multi-output support vector (M-SVR) algorithm. (a) The schematic workflow of the vision-aided location model. (b) Training results of the M-SVR model with 33 sets of data points, evaluated by the parameters of R^2 score, root mean square error (RMSE), mean absolute

error (MAE) and RSME/MAE. (c) Prediction results of the M-SVR model with test data from additional eight independent volunteers, showing a high prediction accuracy.

References

1. Zhang, F., *et al.* Mediapipe hands: On-device real-time hand tracking. *arXiv preprint arXiv:2006.10214*, (2020).
2. Sánchez-Fernández, M., de-Prado-Cumplido, M., Arenas-García, J. & Pérez-Cruz, F. SVM multiregression for nonlinear channel estimation in multiple-input multiple-output systems. *IEEE Trans. Signal Process.* **52**, 2298-2307 (2004).

Modification to the manuscript:

i) On pages 20 and 21, we changed “By employing a monocular camera mounted on the robotic hand, the location information of the pulse-diagnosis point (\mathcal{S}) can be accurately predicted, utilizing a combination of the coordinate transformation of monocular imaging and an image processing technique (see details in Supplementary Fig. S23 and Note S3).”

ii) We changed Supplementary Note S1 of the original supplementary information as “

Supplementary Note S3: Vision-assisted location of pulse-diagnosis point

It is crucial to emphasize that accurate location of the artery pulse point is vital for the autonomous pulse-diagnosis robotic system. The location data of the pulse-diagnosis point is obtained through using an image processing method. A monocular camera is mounted on the robotic hand, and each eye-in-hand capture is firstly fed to the trained hand detector module to determine the hand-knuckle points in the pixel coordinate system (PCS)⁴, marked with solid green dots in Supplementary Fig.

S23a. To better describe the position of the artery pulse point in the image, a hand coordinate system (HCS) is established, using the ring finger metacarpal (MCP) point as the origin. Hence, the transformation matrix (${}^p\mathbf{M}_h$) from HCS to PCS can be expressed by

$${}^p\mathbf{M}_h = \begin{bmatrix} \cos \theta & -\sin \theta & x_{\text{ring_finger_mcp}}^p \\ \sin \theta & \cos \theta & y_{\text{ring_finger_mcp}}^p \\ 0 & 0 & 1 \end{bmatrix}, \quad (\text{S1})$$

and θ is the rotation angle between the coordinate systems and can be calculated by

$$\theta = -\arccos \left(\frac{x_{\text{pinky_mcp}}^p - x_{\text{ring_finger_mcp}}^p}{\| \mathbf{S}_{\text{pinky_mcp}}^p - \mathbf{S}_{\text{ring_finger_mcp}}^p \|} \right), \quad (\text{S2})$$

where $\mathbf{S}_{\text{pinky_mcp}}^p$ and $\mathbf{S}_{\text{ring_finger_mcp}}^p$ are the coordinates of the pinky metacarpal and ring finger metacarpal points in PCS, respectively. Subsequently, multi-output support vector regression (M-SVR), capable of outputting multiple predicted values together, is utilized to predict the position of the pulse point (\mathcal{S}) in HCS⁵. Four key points, specifically the wrist, thumb carpometacarpal (CMC), index finger metacarpal, and middle finger metacarpal, serve as the inputs (\mathbf{X}) of the algorithm, which can be expressed by

$$\mathbf{X} = {}^p\mathbf{M}_h^{-1} \left[\mathbf{S}_{\text{pinky_mcp}}^p, \mathbf{S}_{\text{ring_mcp}}^p, \mathbf{S}_{\text{middle_mcp}}^p, \mathbf{S}_{\text{index_mcp}}^p \right]. \quad (\text{S3})$$

As a result, the pulse point is given as the output of the algorithm, which is then converted into the position in the PCS by the transformation matrix (${}^p\mathbf{M}_h$) and marked with a magenta dot in the image. Following the linear transformation to convert the pulse point into the position in the robot's base coordinate system (BCS), the computer directs the robot hand to the right position for pulse detection.

A total of 33 samples are fed to train the vision-aided location prediction model based on the M-SVR algorithm. The fitting results are presented in Supplementary Fig. S23a and b, where the

coefficient of determination (R^2 score) is calculated as 0.984 (preferred to be close to 1), and the root mean square error (RMSE) and the mean absolute error (MAE) are relatively small as 10.469 pixels and 7.739 pixels, respectively. Moreover, the ratio of RMSE to MAE, equal to 1.353, is close to $\sqrt{\pi/2}$ (≈ 1.253), suggesting that the fitting errors follow a normal distribution. From the training data in Supplementary Fig. S23a, the predicted pulse point (marked with a magenta circle) is closely consistent with the observed pulse point (marked with a blue circle), with specific coordinate values depicted in Supplementary Fig. S23b. These results show that the prediction model has a high goodness of fit. To further validate the prediction accuracy of this model, hand images with different gestures and/or arm sizes from an additional eight independent volunteers are processed by the trained model. The results (bottom of Supplementary Fig. S23a and Fig. S23c) indicate that the predicted pulse-diagnosis point closely accords with the observed value. The RMSE and MAE are exceptionally small and calculated as only 6.314 pixels and 3.875 pixels, respectively. These results highlight the capability of the vision-aided location prediction model to accurately predict the pulse position.”

iii) We also added **Fig. R2** as **Supplementary Fig. S23** in the revised supplementary information.

iv) We replaced the schematic illustration of the hand coordinate system, which is defined with the ring finger metacarpal point serving as the origin, in **Fig. 5a** of the original manuscript by **Fig. R3**.

Fig. R3. Revised Fig. 5a.

Responses to comments of Referee #2

Summary Comment: “In this paper, the authors present a Marangoni-driven deterministic formation approach for fabricating hollow structured microarrays while exploring the influences of significant process parameters. The mechanical properties of hollow microstructures were evaluated through experiments and numerical simulation. The hollow microstructural film can serve as the dielectric layer to significantly enhance the sensitivity and detect limit of capacitive tactile sensors, which has wide potential for practical applications, such as robot tactile. The proposed approach for producing hollow microstructures is scalable and suitable for flexible, large-area microstructures and sensors manufacturing, which is a significant contribution to the field. I think it is more than appropriate for publication in Nature Communication after minor revision.”

Reply: We thank the referee for the positive comments and the recommendation to publish with a minor revision. We have carefully addressed the issues detailed below and revised our manuscript substantially.

Comment 1: “The microstructures are widely utilized in Flexible tactile sensors, which can be produced in various methods. In terms of applications, the authors should provide better discussion and comparison of the methods for creating hollow microstructures and Marangoni-driven formation approaches.”

Reply: We thank the referee for the valuable comment. The discussion and comparison of various fabrication methods will help to enhance the applicability of our approach.

Various attempts have been made to develop different 3D microstructures, such as micro-pillar arrays, micro-domes, micro-stripes, and plant-based irregular structures, through replicating from a premade mold ¹, and electrically responsive self-growing strategy ². Currently, the fabrications of hollow microstructures are studied less to an extent. A hot-pressed method is reported to develop the hollow micro-dome array with a radius of $\sim 750\ \mu\text{m}$ and a height of $\sim 300\ \mu\text{m}$ ³. However, this method is cumbersome due to the preparation conditions of high pressure (5 MPa), heating temperature (75 °C), and stainless-steel masks with uniform concave patterns, which inhibits the application in the preparation of hollow microscale structures. Recently, a wind gun-assisted replica-molding technology has been employed to prepare a graded hollow ball arch structure with a diameter of 2 cm ⁴. A hollow ball arch structure is produced when the wind gun promotes the solution in the mold pit to flow out from the center. However, the wind gun needs to be aligned with the mold pits, which limits the use of this approach to prepare hollow structures of centimeter scale, rather than at the micrometer scale. It would be complicated and challenging to ensure microstructured arrays homogeneity.

In contrast, the Marangoni-driven deterministic formation approach proposed in this paper has the advantages of facilitation, scalability, and large-area compatibility, which can be easily used to prepare microscale hollow structures, such as the hollow micro-pyramid with a base width of $39\ \mu\text{m}$, as shown in Supplementary Fig. S4 of the original supplementary information. Additionally, the shapes of the interior cavity in the microstructured arrays are consistent, where the height of the interior cavity can be tailored by regulating the solution volume and curing temperature as discussed in Fig. 2 of the original manuscript. This method can also be extended to prepare microstructures of other

shapes, such as hollow micro-domes, cylinders, rectangles, etc., as shown in Supplementary Fig. S13 of the original supplementary information and Fig. R4.

Fig. R4. SEM images of the top and bottom surfaces of (a) the hollow cylinder microstructure array, and (b) the hollow cuboid microstructure array. Scale bar, 200 μm .

References

1. Yuan, Y. m., *et al.* Microstructured Polyelectrolyte Elastomer-Based Ionotronic Sensors with High Sensitivities and Excellent Stability for Artificial Skins. *Adv. Mater.* **36**, 2310429 (2024).
2. Tian, H., *et al.* Core-shell dry adhesives for rough surfaces via electrically responsive self-growing strategy. *Nat. Commun.* **13**, 7659 (2022).
3. Chen, S., *et al.* Noncontact heartbeat and respiration monitoring based on a hollow microstructured self-powered pressure sensor. *ACS Appl. Mater. Interfaces* **10**, 3660-3667 (2018).
4. Ding, Z., *et al.* Highly Sensitive Ionotronic Pressure Sensor with Side-by-Side Package Based on Alveoli and Arch Structure. *Adv. Sci.*, 2309407 (2024).

Modification to the manuscript:

i) On page 3, we changed “Various attempts have been made to develop different 3D microstructures, such as micro-pillar arrays ³³, micro-domes ^{6, 34}, micro-stripes ^{29, 35}, and plant-based irregular structures ^{22, 36}, through replicating from a premade mold ³⁷, and electrically responsive self-growing strategy ³⁸.”

ii) On page 4, we added “In comparison, hollow structures, generally containing an interior cavity, will be softer and provide more space for compression under high pressure, which denotes a promising trend for pressure sensors to address the longstanding challenge of high sensitivity over a broad pressure range. Previous work has reported on the preparation of hollow micro-domes, nevertheless, the process of which is cumbersome and challenging to fabricate hollow structured arrays with micrometer scale ⁴⁷.”

iii) We also added **Fig. R4a and b** as **Supplementary Fig. S3a and b** in the revised supplementary information, respectively.

Comment 2: “The performance of the prepared sensor is significantly impacted by the shape of microstructures. The Figure 1b-d shows the shapes of hollow pyramid microstructures. Whether the shapes of these structures are consistent. Can the approach be applied to forming different hollow microstructures?”

Reply: We thank the referee for this comment. The shapes of the interior cavity of the microstructured arrays formed by the Marangoni-driven microfluidic assembly approach are consistent. And the approach can be applied to forming different hollow microstructures.

During the evaporation process of polymer solution, the mold coated with the polymer solution is placed in a horizontal incubator, which ensures a uniform and steady air atmosphere. Here, the surface topography of the interior cavity of the micro-pyramid is measured by using laser scanning confocal microscopy (LSCM, KEYENCE, VK-X200K) at five areas (center and four corners) of the microstructured array (a size of 12.5 mm \times 12.5 mm). The results are shown in Fig. R5, indicating that the shapes and dimensions of the hollow micro-pyramid structures at different locations are consistent. The base width of the micro-pyramid is 125 μm and the solution volume is 1 mL. From the profile curves, the height of the interior cavity is approximately 27 μm , which closely accords with the result measured from the cross-sectional SEM image (Supplementary Fig. S6 of the original supplementary information).

Fig. R5. Surface topography of the interior cavity of the micro-pyramid at five areas (center and four corners) of the microstructured array.

The developed Marangoni-driven microfluidic assembly method is not limited to producing hollow micro-pyramid structures with different sizes (Fig. R6a and d), but can also be extended to fabricate hollow microstructures with various shapes, including but not limited to cylinders, cuboids, and domes. The hollow cylinder and cuboid microstructures are prepared by utilizing SU-8 templates, as shown in Fig. R6b and c. SEM images demonstrate that the top surfaces of the microstructured

films maintain the shapes of the cylinder and cuboid templates, respectively, while the bottom surfaces exhibit a uniform concave configuration. Additionally, the hollow micro-dome structure in Fig. R6e is fabricated by applying a 3D-printing template. Therefore, with the advantages of facilitation, scalability, and large-area compatibility, the proposed method can be adapted to produce hollow microstructured arrays with various shapes by modifying the templates shapes.

Fig. R6. SEM images of the top and bottom surfaces of (a) the hollow micro-pyramid structure array, (b) the hollow micro-cylinder structure array, and (c) the hollow micro-cuboid structure array.

Cross-sectional SEM images of (d) the hollow micro-pyramid structure and **(e)** the hollow micro-domo structure with a base width of 1 mm. Scale bar, 200 μm .

Modification to the manuscript:

i) On pages 7 and 8, we changed “These results demonstrate that the proposed approach is effective to form hollow pyramid microstructures. The surface topographies of the interior cavities of the micro-pyramids are compared at five areas (center and four corners) of the microstructured array (a size of 12.5 mm \times 12.5 mm) by using laser scanning confocal microscopy, indicating consistent shapes and dimensions of the hollow micro-pyramid structures across various locations (Supplementary Fig. S2). The replica molding incorporating with the evaporation-induced microfluidic formation provides a versatile approach to create interior cavities at micrometer scale and can also be applied to develop a wide variety of hollow microstructures with various shapes, no requiring additional processing steps. Supplementary Fig. S3 demonstrates SEM images of the hollow cylinder, cuboid, and dome microstructures that are prepared with the various template shapes.”

ii) We also added **Fig. R5** and **Fig. R6b-e** as **Supplementary Fig. S2** and **Fig. S3** in the revised supplementary information, respectively.

iii) We also changed **Supplementary Fig. S13a and b** of the original supplementary information as **Supplementary Fig. S3c and d**.

Comment 3: “The formation of the sensor is important for practical applications. What are the roles of the different layers of hollow micro-pyramid enhanced CPS sensors, such as the PVA layer and PDMS layer in Figure 1f?”

Reply: We thank the referee for this comment and agree that the formation of the sensor is important for practical applications.

As shown in Fig. 1f of the original manuscript, the developed capacitive pressure sensor consists of five layers: a bottom PVA film with AgNWs electrode, a PDMS insulation layer, a hollow micro-pyramid array (PVA/MgCl₂) employed as the dielectric layer, a top PVA film with AgNWs electrode, and a PDMS encapsulation layer. The bottom undoped PVA film serves as the substrate of capacitive pressure sensor. The AgNWs solution is sprayed and patterned onto the bottom PVA film as the bottom electrode of a capacitor. Subsequently, a thin insulating layer (<10 μm) is created by spin-coating a PDMS mixture onto the PVA substrate. This insulating layer can prevent ionic conduction between the AgNWs electrode and PVA/MgCl₂ dielectric layer to enhance the sensor reproducibility¹. The top electrode layer is identical to the bottom electrode layer, also composed of PVA and AgNWs. Differently, a thick PDMS layer (~230 μm) is applied on the top PVA film to serve as the encapsulation layer, with providing environmental isolation and physical protection.

References

1. Li, Z., *et al.* Gelatin Methacryloyl-Based Tactile Sensors for Medical Wearables. *Adv. Funct. Mater.* **30**, 2003601 (2020).

Modification to the manuscript:

i) On page 8, we added “This CPS consists of five layers, a bottom PVA film with silver nanowires (AgNWs) electrode as the substrate, a PDMS insulation layer, a hollow micro-pyramid array (PVA/MgCl₂) employed as the dielectric layer, a top PVA film with AgNWs electrode, and a PDMS encapsulation layer.”

ii) On page 24, we changed “This PDMS insulating layer prevents ionic conduction between the AgNWs electrode and PVA/MgCl₂ film and enhances the sensor reproducibility⁶². Differently, for the top electrode layer, the spin-coating speed is 300 rpm to obtain a thick PDMS layer (~230 μm) that serves as an encapsulation layer of the sensor, providing environmental isolation and physical protection.”

Comment 4: “The theoretical model would be helpful for the design of the tactile sensor. The scaling law is good. But more details of scaling law of the equivalent compressive stiffness should be included in this paper, particularly about the definition of “*m*” in Figure 3e.”

Reply: We thank the referee for this valuable comment. We agree that it is necessary to clarify the derivation of the scaling law and added detail descriptions of scaling law of the equivalent compressive stiffness and the definition of “*m*” in the revised manuscript.

The compressive behavior of the hollow micro-pyramid structure is investigated through finite element analysis (FEA) simulations to establish the relationship between equivalent compressive stiffness (*k*) and material (Young’s modulus, *E*) and geometric parameters (hollow-to-total height ratio *h/H*, base width *B*, and film thickness *t*), denoted as $k = f(h/H, B, E, t)$. Eight-node solid elements

(C3D8H) were used to model the micro-pyramid structure. The results are illustrated in Supplementary Fig. S9 of the original supplementary information, where the dots represent the raw data and the solid lines are the fitted curves.

The equivalent compressive stiffness significantly decreases by 3.6 times as the height ratio increases from 0.05 to 0.6, where the fitted curve is described as $k \propto (1+h/H)\ln(h/H)$. Similarly, when the base width increases from 45 μm to 150 μm , resulting in a larger interior cavity, the equivalent compressive stiffness decreases by 1.6 times. The relationship can be effectively fitted as $k \propto 1/B$. In Supplementary Fig. S9c of the original supplementary information, a distinct linear relationship can be observed between the compressive stiffness and the Young's modulus of the dielectric film (i.e., $k \propto E$). A decrease of Young's modulus from 500 MPa to 0.5 MPa results in the linear reduction of compressive stiffness by 1000 times, which highlights the significance of doping the inorganic salt MgCl_2 into PVA for lower modulus. The film thickness of the microstructured film varies with the solution volume and curing temperature as shown in Fig. 2k of the original manuscript. While the film thickness increases from 3 μm to 18 μm , the compressive stiffness slightly increases with a linear relationship expressed as $k \propto t$. It is noted that the top width of the pyramid is much less than the base width according to the SEM image in Fig. R7 and previous work¹, illustrating that the actual shape of micro-pyramid structure is typically represented as a frustum. Here, m is denoted as the top width of the micro-pyramid and is approximately equal to 1 μm .

The scaling law can be derived by synthesizing the previously described curve-fitting relationships between the equivalent compressive stiffness and four key parameters of the micro-pyramid structure.

Fig. R7. Side-view SEM image of the micro-pyramid. Scale bar, 10 μm . The inset shows the zoom-in micro-pyramidal tip structure. Scale bar, 2 μm .

References

1. Ruth, S. R. A., *et al.* Rational design of capacitive pressure sensors based on pyramidal microstructures for specialized monitoring of biosignals. *Adv. Funct. Mater.* **30**, 1903100 (2020).

Modification to the manuscript:

i) On page 14, we changed “It is noted that the top width is much less than the base width according to the SEM image in Supplementary Fig. S12, illustrating that the actual shape of micro-pyramid structure is typically represented as a frustum. A scaling law of the equivalent compressive stiffness can be derived as $k \propto Em(0.5\tan\theta+t/B)(1+h/H)\ln(h/H)$ (see details in Fig. 3e and Supplementary Note S1), where m denotes the top width of the pyramid structure.”

ii) On page 38, we changed “(e) Scaling law of the equivalent compressive stiffness as a function of the combined parameter $Em(0.5\tan\theta+t/B)(1+h/H)\ln(h/H)$.”

iii) On page 17 of the revised supplementary information, we added “The dots represent the raw data, and the solid lines are the fitted curves. The fitted curves can be described as $k \propto (1+h/H)\ln(h/H)$, $1/B$, E , and t .”

iv) We added Supplementary Note S1 in the revised supplementary information “

Supplementary Note S1: Derivation of the scaling law for the equivalent compressive stiffness

The compressive behavior of the hollow micro-pyramid structure is investigated through finite element analysis (FEA) simulations to establish the relationship between equivalent compressive stiffness (k) and material (Young’s modulus, E) and geometric parameters (hollow-to-total height ratio h/H , base width B , and film thickness t), denoted as $k = f(h/H, B, E, t)$. Eight-node solid elements (C3D8H) were used to model the micro-pyramid structure. The results are illustrated in Supplementary Fig. S11, where the dots represent the raw data and the solid lines are the fitted curves.

The equivalent compressive stiffness significantly decreases by 3.6 times as the height ratio increases from 0.05 to 0.6, where the fitted curve is described as $k \propto (1+h/H)\ln(h/H)$. Similarly, when the base width increases from 45 μm to 150 μm , resulting in a larger interior cavity, the equivalent compressive stiffness decreases by 1.6 times. The relationship can be effectively fitted as $k \propto 1/B$. In Supplementary Fig. S11c, a distinct linear relationship can be observed between the compressive stiffness and the Young's modulus of the dielectric film (i.e., $k \propto E$). A decrease of Young's modulus from 500 MPa to 0.5 MPa results in the linear reduction of compressive stiffness by 1000 times, which highlights the significance of doping the inorganic salt magnesium chloride (MgCl_2) into polyvinyl alcohol (PVA) for lower modulus. While the film thickness increases from 3 μm to 18 μm , the compressive stiffness slightly increases with a linear relationship expressed as $k \propto t + 0.5B\tan\theta$, where

$B = 60 \mu\text{m}$ and $\theta = 54.7^\circ$. With the baseline values $E = 2 \text{ MPa}$, $B = 60 \mu\text{m}$, $t = 8 \mu\text{m}$, and $h/H = 0.2$, variation of any of the parameters in $f(h/H, B, E, t)$ yields approximately the same straight line for the relationship between the equivalent compressive stiffness and four key parameters of the micro-pyramid structure.”

vi) We also added **Fig. R7** as **Supplementary Fig. S12** in the revised supplementary information.

Comment 5: “In Figure 4b, the authors mention that the low limit of detection of the prepared sensor is 0.21 Pa. Can it be decreased further in the future?”

Reply: We thank the referee for this useful comment. We believe that the limit of detection of the prepared sensor can continue to be improved in the future. The reasons are discussed below.

The detection limit of capacitive pressure sensor is usually in accordance with its sensitivity, which is influenced by the compressibility of the dielectric layer. In section 2.3 of the original manuscript, we discussed the compressive behavior of the proposed hollow micro-pyramid structure. The equivalent compressive stiffness (k) is closely related to geometric and material parameters of the micro-pyramid structure: the hollow-to-total height ratio (h/H), base width (B), and Young's modulus (E), as shown in Supplementary Fig. S9 of the original supplementary information. In addition to the compressive performance of a single micro-pyramid structure, the distribution of hollow microstructures also plays a crucial role in determining the overall equivalent compressive stiffness of capacitive sensor, as discussed in previous works¹⁻³. Hence, we believe that the following three strategies might be explored to improve the detection limit by enhancing structural compressibility.

i) A lower equivalent compressive stiffness is possible by spreading out the microstructures to raise the air-to-elastomer ratio (Fig. R8a), due to the ineffectiveness of air in resisting deformation, which is favorable for a higher sensitivity. From the FEA results in Fig. R8b and c, it is observed that as the interval space (d) between the pyramid structures increases from B to $3B$, the compressive strain proportionally increases under the same applied pressure. Moreover, this trend will become more apparent when subjected to a larger pressure. The experimental data in Fig. R9 indicates that the capacitive pressure sensor exhibits higher sensitivity when utilizing a microstructure arrayed dielectric film with a larger interval space. The spacing between hollow micro-pyramids employed in Fig. 4 of the original manuscript is equal to the base width of the micro-pyramid. Therefore, increasing the interval space to a value of $2B$ or $3B$ will effectively enhance the detection limit of the pressure sensor.

ii) Enlarging the interior cavity (e.g., increasing height ratio) of micro-pyramid structures will result in a significant reduction of the equivalent compressive stiffness, as illustrated in Fig. R8d. This conclusion is further corroborated by the experimental results presented in Fig. 4a of the original manuscript, depicting the relative capacitance change versus pressure. As discussed in section 2.2 of the original manuscript, the height ratio is determined by two critical parameters: solution volume and curing temperature. The microstructured dielectric layer utilized for testing in Fig. 4b of the original manuscript is prepared under the conditions of a solution volume of 0.5 mL and room temperature curing. Consequently, it would be beneficial to reduce the solution volume or increase the curing temperature to achieve a larger cavity.

iii) An additional strategy is reducing the Young's modulus of the microstructured dielectric film. As demonstrated in Fig. R8e, the equivalent compressive stiffness of the hollow micro-pyramid

structure is correlated with Young's modulus. By incorporating the inorganic salt $\text{MgCl}_2 \cdot 6\text{H}_2\text{O}$ into PVA with weight ratios from 20% to 100%, the modulus of modified PVA film significantly decreases from 2.6 MPa to 0.12 MPa, as shown in Fig. 3g of the original manuscript. The microstructured dielectric layer with a weight ratio of 20% is employed in the fabrication of the capacitive pressure sensor in Fig. 4b of the original manuscript. Therefore, increasing the doping ratio of MgCl_2 would result in a softer microstructured dielectric film, such as a weight ratio of 40% yielding a modulus of 0.92 MPa.

Fig. R8. Compressive strain analysis of the hollow micro-pyramid structures under different applied pressures by the finite element method. (a) Schematic illustration of the distribution of hollow microstructures, including base width (B) and interval space (d). (b) FEA results of compressive deformation of the micro-pyramid structures with various interval spaces under different levels of applied pressure (0.5 kPa, 5 kPa, and 20 kPa). (c) Contour plot of the compressive strain of the micro-pyramid structure in terms of applied pressure and d/B for $h/H = 0.3$ and $E = 2.6$ MPa. (d)

Contour plot of the compressive strain of the micro-pyramid structure in terms of applied pressure and h/H for $d/B = 1$ and $E = 2.6$ MPa. (e) Contour plot of the compressive strain of the micro-pyramid structure in terms of applied pressure and E for $h/H = 0.2$ and $d/B = 1$.

Fig. R9. Pressure response of relative capacitance change of the developed capacitive pressure sensor with different interval spaces.

References

1. Wan, Y., *et al.* A Highly Sensitive Flexible Capacitive Tactile Sensor with Sparse and High-Aspect-Ratio Microstructures. *Adv. Electron. Mater.* **4**, 1700586 (2018).
2. Ruth, S. R. A., *et al.* Rational design of capacitive pressure sensors based on pyramidal microstructures for specialized monitoring of biosignals. *Adv. Funct. Mater.* **30**, 1903100 (2020).
3. Ruth, S. R. A. & Bao, Z. Designing tunable capacitive pressure sensors based on material properties and microstructure geometry. *ACS Appl. Mater. Interfaces* **12**, 58301-58316 (2020).

Modification to the manuscript:

- i) On page 17, we added “The detection limit of developed sensor is in accordance with its sensitivity, which can be further optimized as discussed in Supplementary Note S2.”
- ii) We added Supplementary Note S2 in the revised supplementary information “

Supplementary Note S2: Performance optimization of the hollow micro-pyramid enhanced pressure sensor

The sensing performance of the hollow micro-pyramid (HMP) enhanced capacitive pressure sensor (CPS) is influenced by its compressibility of the dielectric layer, which is closely related to geometric and material parameters: interval space to base width (d/B), base width (B), hollow-to-total height ratio (h/H), and Young's modulus (E), as discussed in previous works¹⁻³. The compressive behavior of the microstructured dielectric layer is investigated through finite element analysis (FEA) simulations, as shown in Supplementary Fig. S19.

First, from the FEA results in Supplementary Fig. S19b and c, it is observed that as the interval space (d) between the pyramid structures increases from B to $3B$, the compressive strain proportionally increases under the same applied pressure. The experimental data in Supplementary Fig. S20a indicates that the CPS exhibits higher sensitivity when utilizing a microstructure arrayed dielectric film with a larger interval space. Therefore, increasing the interval space to a value of $2B$ or $3B$ will effectively enhance the sensitivity of the pressure sensor.

Second, Supplementary Fig. S20b demonstrates that the reduction of the pyramid size will be more favorable to obtain a higher sensitivity, and this conclusion has also been verified in other works (Supplementary Table S1). The sensor based on the micro-pyramid with a smaller base width of $39\ \mu\text{m}$ exhibits a higher sensitivity. However, the interior cavity within a smaller micro-pyramid can be easily collapsed under pressure, limiting the high sensitivity to a small pressure range, as depicted in Supplementary Fig. S20b. Therefore, a hollow micro-pyramid with a base width of $79\ \mu\text{m}$ was selected as the final sensor for testing in this paper.

Third, enlarging the interior cavity (e.g., increasing height ratio) of micro-pyramid structures will result in a significant increment of the compressive strain, as illustrated in Supplementary Fig. S19d. This conclusion is further corroborated by the experimental results presented in Fig. 4a of the manuscript. The height ratio is determined by two critical parameters: solution volume and curing temperature. The microstructured dielectric layer utilized for testing in Fig. 4 of the manuscript is prepared under the conditions of a solution volume of 0.5 mL and room temperature curing. Consequently, it would be beneficial to reduce the solution volume or increase the curing temperature to achieve a larger cavity.

Fourth, the compressive strain rises linearly as the modulus of modified PVA film significantly decreases (Supplementary Fig. S19e). The experiment results in Supplementary Fig. S18 present that the CPS based on the modified PVA (Young's modulus of 2.6 MPa) film with a doping ratio of 20% exhibits a higher sensitivity compared with the sensor based on the pure PVA film (Young's modulus of 839 MPa). As the doping ratio of MgCl_2 increases, the polymer film becomes softer, but the elastic recovery also begins to deteriorate. The modified PVA film with a doping ratio of 20%, serving as the dielectric layer of the CPS in this paper, exhibits favorable rubber-like mechanical properties, with Young's modulus of 2.6 MPa and elastic recovery rate of 99% under a tensile strain of 50% (Supplementary Fig. S15). At a doping ratio of 40%, the Young's modulus declines to 0.92 MPa with an elastic recovery of 96%. With a further increase to 60%, the elastic recovery ratio decreases to below 90% with a lower modulus of 0.63 MPa. Therefore, applying the film with a doping ratio of 40% would be a considerable strategy to achieve a higher sensitivity or a lower detection limit."

iii) We also added **Fig. R8** and **Fig. R9** as **Supplementary Fig. S19** and **Fig. S20a** in the revised supplementary information.

Comment 6: “There is an inconsistency between the references in **Figure 4c** and the references in the main body of the manuscript.”

Reply: We thank the referee for pointing out this issue. We examined and revised reference numbers in Fig. 4c of the original manuscript as shown in Fig. R10, in accordance with references in the main body of the manuscript.

Fig. R10. Revised Fig. 4c

Modification to the manuscript: We revised the number of references and replaced **Fig. 4c** of the original manuscript with **Fig. R10**.

Comment 7: “In **Figure 5a**, the definition of **ARM**, **DDR3**, and **UDP** needs to be explained clearly.”

Reply: We thank the referee for pointing out this issue. We added the relevant content in the revised manuscript to explain **ARM**, **DDR3**, and **UDP**. **ARM** is an abbreviation for **Advanced RISC Machine**, and represents a family of reduced instruction set computing (**RISC**) architectures designed

for computer processors. DDR3 stands for Double Data Rate Type 3, which is a type of synchronous dynamic random access memory (SDRAM) used for system memory. UDP is short for User Datagram Protocol, which serves as a transport layer protocol utilized to transmit the acquired capacitance data to the computer during the demonstration.

Modification to the manuscript:

On page 42, we added “Advanced RISC Machine (ARM) is a family of reduced instruction set computing (RISC) architectures designed for computer processors, Double Data Rate Type 3 (DDR3) is a type of synchronous dynamic random access memory used for system memory, and User Datagram Protocol (UDP) is transport layer protocol.”

Responses to comments of Referee #3

Summary Comment: “The manuscript titled "Marangoni-driven deterministic formation of softer, hollow microstructures for sensitivity-enhanced tactile system" reported a wearable capacitive tactile sensor based on a hollow micro-pyramid microstructure. The controllable drying process of the polymer solution rendered adjustable interior cavity properties, high sensitivity, and an extensive measurement range. The authors tried to present a comprehensive understanding of the working mechanism, encompassing fundamental sciences and practical applications. Generally, this manuscript exhibits good organization, logical clarity, and a balanced integration of experiments and theories. However, specific concerns require attention, which are outlined below.”

Reply: We thank the referee for the constructive comments that help to improve our manuscript substantially. We have carefully addressed the issues detailed below and revised our manuscript accordingly.

Comment 1: “What standards define the base width range of hollow microstructures? The size and distribution of microstructures significantly impact sensitivity and detection range, and the author should furnish evidence supporting the optimization of the current selection. Additionally, if different sizes and distributions of microstructure combinations can achieve superior results?”

Reply: We thank the referee for this comment. We are sorry for unclearly defining the base width of hollow microstructures in the original manuscript. As depicted in Fig. R11 and Fig. 2a of the original manuscript, the base width (B) is determined as the bottom length of the outer surface of a micro-pyramid structure, which is the same as that of a solid pyramid.

Fig. R11. Definition of the base width (B) of a hollow micro-pyramid structure. (a) Schematic illustration and (b) top-view SEM image of the hollow micro-pyramid structure. Scale bars, 200 μm . The inset shows the zoom-in micro-pyramid structure. Scale bar, 30 μm .

We agree that the size and distribution of microstructures are important for sensor performance. First, it is obvious that a lower equivalent compressive stiffness is possible by spreading out the microstructures to rise the air-to-elastomer ratio, due to the ineffectiveness of air in resisting deformation, which is favorable for higher sensitivity. Figure R12a presents the pressure response of relative capacitance change of the HMP-enhanced sensor with different ratios of interval space (d) to base width ($d/B = 1, 2, \text{ and } 3$). A larger spacing (i.e., $d/B = 3$) exhibits higher sensitivity, representing a simple and universal strategy applicable to all microstructure-based capacitive pressure sensors. Notably, after reviewing the literature on micro-pyramid array-based capacitive sensors (Supplementary Table S1 of the original supplementary information), the performance of the sensor based on the microstructured dielectric film with $d/B = 1$ are further explored in this paper for the convenience of comparing its performance with previous studies. Second, Figure R12b demonstrates that the reduction of the pyramid size will be more favorable to obtain a higher sensitivity, and this conclusion has also been verified in other works, which have been collected and collated in Supplementary Table S1 of the original supplementary information. The sensor based on the micro-

pyramid with a smaller base width of 39 μm exhibits a higher sensitivity. However, the interior cavity within a smaller micro-pyramid can be easily collapsed under pressure, limiting the high sensitivity to a small pressure range, as depicted in Fig. R12b. Therefore, a hollow micro-pyramid with a base width of 79 μm was selected as the final sensor for testing in this paper.

Fig. R12. Relative capacitance change of the hollow micro-pyramid enhanced sensor with different (a) normalized distributions (d/B), and (b) base widths (B).

Modification to the manuscript:

i) On page 9, we added “The base width (B) is determined as the bottom length of the outer surface of a micro-pyramid structure, which is the same as that of a solid pyramid.”

ii) On pages 16 and 17, we changed “Additionally, the sensing performances of the HMP-enhanced CPS, relative to different material and geometric parameters, have been elaborated in detail (Supplementary Note S2). Young’s modulus of the structured dielectric film, as another critical feature that affects the compressibility of the pressure sensor, has also been investigated to improve the sensing performance.”

iii) On page 17, we added “A sparser distribution by spreading out the microstructures to increase the air-to-elastomer ratio, due to the ineffectiveness of air in resisting deformation, is

favorable for higher sensitivity, which represents a simple and universal strategy applicable to all microstructure-based CPSs (Supplementary Fig. S19a-c and Fig. S20a). A smaller pyramid will be more favorable to obtain a higher sensitivity, which has been verified in other works (Supplementary Table S1). However, the interior cavity within a smaller micro-pyramid can be easily collapsed under pressure, limiting the high sensitivity to a small pressure range (Supplementary Fig. S20b).”

iv) We added Supplementary Note S2 in the revised supplementary information “

Supplementary Note S2: Performance optimization of the hollow micro-pyramid enhanced pressure sensor

The sensing performance of the hollow micro-pyramid (HMP) enhanced capacitive pressure sensor (CPS) is influenced by its compressibility of the dielectric layer, which is closely related to geometric and material parameters: interval space to base width (d/B), base width (B), hollow-to-total height ratio (h/H), and Young's modulus (E), as discussed in previous works¹⁻³. The compressive behavior of the microstructured dielectric layer is investigated through finite element analysis (FEA) simulations, as shown in Supplementary Fig. S19.

First, from the FEA results in Supplementary Fig. S19b and c, it is observed that as the interval space (d) between the pyramid structures increases from B to $3B$, the compressive strain proportionally increases under the same applied pressure. The experimental data in Supplementary Fig. S20a indicates that the CPS exhibits higher sensitivity when utilizing a microstructure arrayed dielectric film with a larger interval space. Therefore, increasing the interval space to a value of $2B$ or $3B$ will effectively enhance the sensitivity of the pressure sensor.

Second, Supplementary Fig. S20b demonstrates that the reduction of the pyramid size will be more favorable to obtain a higher sensitivity, and this conclusion has also been verified in other works (Supplementary Table S1). The sensor based on the micro-pyramid with a smaller base width of 39 μm exhibits a higher sensitivity. However, the interior cavity within a smaller micro-pyramid can be easily collapsed under pressure, limiting the high sensitivity to a small pressure range, as depicted in Supplementary Fig. S20b. Therefore, a hollow micro-pyramid with a base width of 79 μm was selected as the final sensor for testing in this paper.

Third, enlarging the interior cavity (e.g., increasing height ratio) of micro-pyramid structures will result in a significant increment of the compressive strain, as illustrated in Supplementary Fig. S19d. This conclusion is further corroborated by the experimental results presented in Fig. 4a of the manuscript. The height ratio is determined by two critical parameters: solution volume and curing temperature. The microstructured dielectric layer utilized for testing in Fig. 4 of the manuscript is prepared under the conditions of a solution volume of 0.5 mL and room temperature curing. Consequently, it would be beneficial to reduce the solution volume or increase the curing temperature to achieve a larger cavity.

Fourth, the compressive strain rises linearly as the modulus of modified PVA film significantly decreases (Supplementary Fig. S19e). The experiment results in Supplementary Fig. S18 present that the CPS based on the modified PVA (Young's modulus of 2.6 MPa) film with a doping ratio of 20% exhibits a higher sensitivity compared with the sensor based on the pure PVA film (Young's modulus of 839 MPa). As the doping ratio of MgCl_2 increases, the polymer film becomes softer, but the elastic recovery also begins to deteriorate. The modified PVA film with a doping ratio of 20%, serving as

the dielectric layer of the CPS in this paper, exhibits favorable rubber-like mechanical properties, with Young's modulus of 2.6 MPa and elastic recovery rate of 99% under a tensile strain of 50% (Supplementary Fig. S15). At a doping ratio of 40%, the Young's modulus declines to 0.92 MPa with an elastic recovery of 96%. With a further increase to 60%, the elastic recovery ratio decreases to below 90% with a lower modulus of 0.63 MPa. Therefore, applying the film with a doping ratio of 40% would be a considerable strategy to achieve a higher sensitivity or a lower detection limit."

v) We added **Fig. R12** as **Supplementary Fig. S20** in the revised supplementary information.

Comment 2: "The authors are recommended to elaborate on achieving precise detection and sensing of local pressure within a confined device area and address the potential influence of microarray electrode preparation on sensitivity and robustness."

Reply: We thank the referee for this important issue. The potential influence of local pressure within a confined device area is very significant for sensing in practical applications.

For localized pressure detection, a 3×3 sensing array with hollow micro-pyramid enhanced capacitive sensors has been constructed, as shown in Fig. R13. In order to detect the crosstalk of the capacitive signals between the pressure sensors, each sensing unit is individually wired with two separate wires that are connected to the LCR meter. The response of sensing array is precisely reflected in corresponding pressure mapping under a cylinder for single input and a H-shaped acrylic plate for multiple input. The results demonstrate a little influence between the sensing units within the array.

Fig. R13. Pressure distribution mapping of the sensing array under (a) a cylinder, and (b) a H-shaped acrylic plate.

When the size of the electrode gap reduces further, the compression of sensing array in one unit will eventually cause the entire top electrode layer to sink, interfering with the surrounding sensing unit. As demonstrated in our previous works, a spacer can be inserted between the top and bottom electrode layers of the sensing array to effectively eliminate the spatial crosstalk between sensing units under external pressure (Fig. R14)¹. Zhang et al. elaborate that the spacer structure produces a strain local confinement effect with reducing the deformation overflow of sensing unit by $\sim 90\%$ compared to that of conventional flexible electronics². This approach effectively enables precise detection of local pressure without interference from other sensing units, such as when measuring wind pressure distribution around an airfoil.

Fig. R14. Capacitive pressure sensor strip with a spacer as the intermediate layer ¹.

References

1. Xiong, W., *et al.* Bio-Inspired, Intelligent Flexible Sensing Skin for Multifunctional Flying Perception. *Nano Energy* **90**, 106550 (2021).
2. Zhang, Y., *et al.* Localizing strain via micro-cage structure for stretchable pressure sensor arrays with ultralow spatial crosstalk. *Nat. Commun.* **14**, 1252 (2023).

Modification to the manuscript:

i) On pages 21 and 22, we added “Additionally, for localized pressure detection, a 3×3 sensing array with HMP-enhanced capacitive sensors has been constructed (Supplementary Fig. S24). The response of sensing array is precisely reflected in corresponding pressure mapping under a cylinder for single input and an H-shaped acrylic plate for multiple input, demonstrating a little influence between the sensor units within the array. For a micro-pressure array, a spacer structure can be inserted between the top and bottom electrode layers of the sensing array to create a strain local confinement effect and effectively eliminate the spatial crosstalk between sensing units under external pressure ^{14, 24}.”

ii) We also added **Fig. R13** as **Supplementary Fig. S24** in the revised supplementary information.

Comment 3: “The author could demonstrate more practical scenarios that span various pressure ranges to highlight the device's sensitivity and robustness advantages and provide practical applications to showcase the versatility of the proposed sensor.”

Reply: We thank the referee for this useful comment. We design and add several experiments to demonstrate the high sensitivity and robustness of the hollow micro-pyramid (HMP) enhanced pressure sensor over a wide pressure range. Subtle pressures are applied on the developed sensor with three different preloads: without preload, with 4 kPa preload, and with 12 kPa preload.

First, the HMP-enhanced sensor without preload was exposed to tiny pressures that are encountered by human skin on a regular basis, such as water droplets and respiration. Fig. R15a presents the response of the HMP-enhanced pressure sensor could differentiate the static pressures induced by three water droplets applied one after another. The weight of a drop of water is approximately 24.2 mg, corresponding to an average pressure of 2.42 Pa. In addition, this HMP-enhanced sensor attached to the inner surface of a mask can be used for monitoring the respiratory status of the human to release early-warning signals. As shown in Fig. R15b, the blue curve shows a slow respiration (~12 times/min) within the frequency range of a normal adult, while the red curve presents a fast breathing, approximately 32 times/min. This sensor is able to distinguish the slow and fast respiration evidently.

The HMP-enhanced pressure sensor can also be attached to the index finger for tactile sensing (e.g., grasping the balloon, as shown in Fig. R15c). When the finger touches the balloon, the capacitance of the developed sensor rises rapidly and then remain a certain value. The initial shape of the balloon is marked with red dotted lines in the inset of Fig. R15c. Then, three consecutive tiny

deflations of the balloon produce three consecutive stepwise decreases in pressure, which is hard to feel by hand, but can be detected by the developed sensor. The zoom-in insets indicate the subtle shape changes of the balloon under three tiny deflations, which are marked with green, purple, and yellow dotted lines, respectively. These results indicate that the HMP-enhanced pressure sensor is capable of detecting small pressure changes under a large preload (about 12 kPa). Additionally, in Fig. 5 of the original manuscript, the sensor attached on a robot hand is also shown to detect a small pulse pressure under a preload of about 4 kPa. All these show that the developed sensor exhibits a high sensitivity over different pressure range, applicable for various practical applications.

Fig. R15. Applications of hollow micro-pyramid enhanced sensor in different practical scenarios. (a) Detection of three successive waterdrops on the developed sensor. (b) Monitoring of human respiration status by the sensor integrated into a mask. (c) Sensing the tiny deflations of a balloon under a large preload of approximately 12 kPa when the sensor attached to the index finger.

Modification to the manuscript:

i) On pages 19 and 20, we added “Figure 4h and i present the applications of HMP-enhanced sensors in various practical scenarios, such as tiny pressures encountered by human skin on a regular basis or large pressures by grasping. In the left of Fig. 4h, this sensor without preload could differentiate the static pressures induced by three waterdrops applied one after another, corresponding to approximately 24.2 mg for a drop of water. When the developed sensor is integrated with the inner surface of a mask, it can be used for monitoring the respiratory status of the human to release early-warning signals, e.g. a slow and rapid respiration of approximately 12 times per minute (blue curves) and 32 times per minute (red curves) in right of Fig. 4h, respectively. Additionally, the HMP-enhanced pressure sensor can also be attached to the index finger for tactile sensing with approximately 12 kPa preload (e.g., grasping the balloon). The initial shape of the balloon is marked with red dotted lines in the inset of Fig. 4i. Then, three consecutive tiny deflations of the balloon produce three consecutive stepwise decreases in pressure, which is hard to feel by hand, but can be detected by the developed sensor. The zoom-in insets illustrate the subtle shape changes of the balloon under three tiny deflations, which are marked with green, purple, and yellow dotted lines, respectively. These experiments demonstrate that the HMP-enhanced pressure sensor has a high sensitivity and robustness over a wide pressure range.”

ii) On page 39, we added “**(h)** Detection of three successive waterdrops and monitoring of human respiration status. **(i)** Sensing the tiny deflations of a balloon under a large preload of approximately 12 kPa when the sensor attached to the index finger.”

iii) We added **Fig. R16** as **Fig. 4h and i** in the revised manuscript.

Fig. R16. The added Fig. 4h and i.

Comment 4: “Please investigate the baseline behavior over extended pulse signal testing periods (refer to Fig. 5l), explore the reasons for baseline monotonic fluctuations, and assess the impact of physical activity on pulse testing in portable wearable systems. Additionally, the authors are recommended to discuss strategies to reconcile device sensitivity with external environmental interference in practical applications.”

Reply: We thank the referee for this valuable comment. The developed sensor based on hollow micro-pyramid exhibits good stability under different levels of external loads, as shown in Fig. R17, where the capacitance signals remain almost horizontal during each step pressure and change without hysteresis.

In the Fig. 5l of the original manuscript, a period of time is spent to establish a steady contact for gentle touch between the sensor and the volunteer's wrist due to the softness of human skin. Consequently, when the index finger of the robot bends and touches the pulse-diagnosis point, the capacitance value of the sensor begins to gradually increase and stabilize after about 10 s. Because the fingers of robot are relatively hard and there is a noticeable amount of contact pressure, it will result in slight discomfort and wobble of volunteer's wrist during the pulse detection, leading to the

baseline monotonic fluctuations of the capacitive signal (the inset of Fig. 5I of the original manuscript). This will take time for a volunteer to adapt to the press from a robotic hand. It is observed that the baseline of the signal becomes stable as the longer the measurement time. Therefore, we can utilize the sequential data of monitoring pulse with larger values of time in practice, which is relatively stable.

Fig. R17. Supplementary Fig. S15 of the original supplementary information.

In practical applications, the operating environment of pressure sensor is usually complicated and changeable. Therefore, it is necessary to discuss the strategies for resisting interference from the external environment.

First, the capacitive pressure sensors are susceptible to external electromagnetic interference due to a low capacitance values (from a few picofarad to tens of picofarad). In the application of robotic pulse diagnostic (Fig. 5 of the original manuscript), a capacitance-to-digital converter (CDC) chip is simultaneously integrated into the robot hand, which is located extremely close to the developed sensor attached to the tip of the index finger, to transform the capacitance value into stable digital signals. The CDC chip near the sensor can greatly improve the resistance of electromagnetic interference, which have been validated in our previous work ¹.

Second, variations in the surrounding temperature and humidity may have an impact on the capacitive value of the sensor. The effect of humidity on the sensor can be effectively prevented from contact with the external environment by encapsulating the sensor (i.e., electrodes, dielectric layers), which have been investigated in our previous work ². As shown in Fig. R18a, for different relative humidity environments (60%, 70% and 80%), it is observed that the developed sensor exhibits an almost uniform response under a weight of 5 g. Additionally, Figure R18b presents the response of developed sensor in the various ambient temperature from 25 °C to 45 °C, which covers the temperature of human body. The difference of relative capacitance variation of developed sensor is little under the same load (5 g).

Fig. R18. The response of the developed sensor under a weight of 5 g with different (a) relative humidity (60%, 70% and 80%), and (b) ambient temperatures (25 °C, 35 °C, and 45 °C).

Reference

1. Xiong, W., *et al.* Bio-Inspired, Intelligent Flexible Sensing Skin for Multifunctional Flying Perception. *Nano Energy* **90**, 106550 (2021).
2. Xiong, W., *et al.* Multifunctional Tactile Feedbacks towards Compliant Robot Manipulations via 3D-shaped Electronic Skin. *IEEE Sens. J.* **22**, 9046-9056 (2022).

Modification to the manuscript :

i) On pages 17 and 18, we changed “The developed sensor based on hollow micro-pyramid exhibits good stability under different levels of external loads, where the capacitance signals remain almost horizontal during each step pressure and change without hysteresis.”

ii) On page 21, we added “Because the fingers of robot are relatively hard and there is a noticeable amount of contact pressure, it will result in slight discomfort and wobble of volunteer's wrist during the pulse detection, leading to the baseline slight fluctuations of the capacitive signal. This will take time for a volunteer to adapt to the press from a robotic hand. It is observed that the baseline of the signal becomes stable as the longer the measurement time.”

iii) On page 19, we added “Since environmental temperature and relative humidity commonly affect the response of capacitive devices, the stability of developed sensors under varying temperatures and humidity is also essential to practical applications. Here, the developed pressure sensor is well packaged, such that the humidity change does not affect the signal (Supplementary Fig. 22a). Supplementary Fig. 22b presents the response of developed sensor in the various ambient temperature from 25 °C to 45 °C, which covers the temperature of human body. The difference of relative capacitance variation of developed sensor is a little under the same load (5 g).”

iv) We also added **Fig. R18** as **Supplementary Fig. S22** in the revised supplementary information.

REVIEWERS' COMMENTS

Reviewer #1 (Remarks to the Author):

The revised manuscript has well addressed the comments, thus, it is acceptable without change now.

Reviewer #2 (Remarks to the Author):

The authors have satisfactorily addressed all the critics raised the last time and I would recommend the acceptance of this paper.

Detailed response to referees' comments

Responses to comments of Referee #1

Summary Comment: “The revised manuscript has well addressed the comments, thus, it is acceptable without change now.”

Reply: We thank the reviewer for the recommendation to publish our manuscript in *Nature Communications* as is.

Responses to comments of Referee #2

Summary Comment: “The authors have satisfactorily addressed all the critics raised the last time and I would recommend the acceptance of this paper.”

Reply: We thank the reviewer for the recommendation to publish our manuscript in *Nature Communications* as is.